# FAFO: Lossy KV Cache Compression for Lossless Inference Acceleration via Draftless Fumble Decoding

## Abstract

*Lossy KV cache compression* is a well-explored subfield of machine learning efficiency, with improved latency being one of its major gains. However, lossy compression techniques can fumble from time to time, exhibiting various — and often catastrophic — failure patterns that are not only difficult to resolve but sometimes even hard to identify in the first place, making the direct deployment of models with compressed KV cache a risky endeavor. In this work, we explore a way to preserve *lossless* generation quality while still benefiting from the acceleration provided by attending only to a compressed KV cache. Specifically, we draw inspiration from the *n-gram candidate pool decoding* paradigm pioneered by Lookahead Decoding — a largely overlooked and underdeveloped way to achieve efficient yet lossless decoding — where we purposely allow the model to Fumble Around with compressed KV cache to generate multiple lossy "n-gram guesses" with just one forward pass, while Find Out via lossless verification in the same forward pass in truly parallel fashion. From a conceptual standpoint, our proposed framework is compatible with all typical static or dynamic KV cache compression methods from the token dropping realm, thus opening up a new avenue for the stagnant n-gram decoding paradigm. Practically, we show that — with careful system support — this framework presents many useful traits that similar draftless baselines (e.g., Self-Speculative Decoding) simply cannot achieve, such as requiring only one set of KV cache and being far less sensitive to model, task, and input-length scenarios. Our comprehensive empirical results show FAFO provides 1.20-2.71$\times$ latency speedup over the original model, while consistently outperforming other lossless + draftless solutions by a large margin.

## 1 Introduction and Background

Transformer-based Large Language Models (LLMs) have demonstrated strong capabilities across a wide range of general and specialized tasks. However, one innate challenge of relying on transformer-based architectures is the Key-Value (KV) cache, which is necessary for efficient inference. The KV cache typically grows linearly with batch size and sequence length, and its sheer size creates a significant efficiency bottleneck for model-serving systems, as noted in prior art like Pope et al. (2023); Fu (2024). For these reasons, *KV Cache Compression* as a subfield has received major interest and advancement over the past year (Yuan et al., 2024; Luohe et al., 2024), with many lossy efficiency methods proposed to allow models to process using only compressed KV cache, thereby achieving significant savings on memory footprint and improved latency.[1] In this work, we present **a framework that maintains lossless generation quality while using lossy KV cache compression as its means** — Fumble Around and Find Out (FAFO) — within the draftless efficient decoding paradigm. Specifically, FAFO leverages a modified version of the n-gram candidate-pool

---

[1]We want to faithfully emphasize that FAFO does not provide savings on memory footprint over full-model inference, but only offers latency advantages. We note that this is the norm for all efficient decoding methods that pursue lossless generation quality, since lossless verification cannot be achieved without full attention over the full KV cache. However, FAFO does present significant memory-footprint savings over typical speculative decoding methods. More on this in Section 2 and Section 3.1.

decoding paradigm pioneered by Lookahead Decoding (Fu et al., 2024) and introduces several significant advancements.

We will show that, while integrating token dropping–based KV cache compression into the draftless and lossless efficient decoding paradigm is conceptually straightforward — as already attempted by various self-speculative decoding works like Self-Speculative (Zhang et al., 2023a), TriForce (Sun et al., 2024), MagicDec (Sadhukhan et al., 2025), and SWIFT (Xia et al., 2025) — its integration with n-gram candidate-pool decoding (hereinafter "n-gram decoding") presents unique system challenges but, when done correctly, yields substantial gains. Specifically, we find that (when implemented with careful system support), **this integration exhibits preferable properties that prior self-speculative decoding methods simply cannot match, such as requiring maintenance of only a single set of KV cache and being agnostically more compatible with different models, tasks, and input sequence lengths.** Further, thanks to the inherent characteristics of n-gram decoding and the advances in FAFO, our framework can produce many more "n-gram guesses" — addressing a major limitation noted by the Lookahead Decoding authors — and is therefore particularly effective for tasks that benefit from repetition, such as summarization, writing improvement, and coding, which are common uses of today's LLMs. As a result, the FAFO framework provides a 1.20–2.71$\times$ latency speedup over the original model under vanilla decoding, presenting a large margin over other draftless solutions that also preserve lossless generation quality.

## 1.1 LOSSY KV CACHE COMPRESSION CAN SOMETIMES FAIL, AND FAIL SO IN UNPREDICTABLE WAYS

One unresolvable challenge of lossy efficiency approaches is they do fail under certain model–method–task–setting–compression rate combinations. Prior benchmark works like Yuan et al. (2024) indicate that when they fail, they often fail catastrophically.[2] Such failure-triggering conditions are often tricky to identify, as they can be hidden under layers of different settings. Another commonly overlooked aspect of lossy compression is that *task accuracy is only a proxy for model utility*. Equally important is *behavioral robustness* — whether the model responds consistently — which compression often undermines. Dutta et al. (2024) shows that even at relatively high bitwidths (W8A16, W8A8), models may retain similar accuracy yet undergo substantial behavioral drift. For instance, after compressing Llama2-13B with SmoothQuant to W8A8 (Xiao et al., 2023), 18.99% of MMLU answers (Hendrycks et al., 2021) flipped from correct to incorrect or vice versa, suggesting that lossy-compressed models may behave markedly differently despite comparable accuracy metrics.

In other words, **while KV cache compression offers substantial efficiency gains, deploying models that rely directly on lossy compressed KV caches often risks reduced reliability.** Given the infinitely diversified task scenarios, service owners cannot feasibly surface all such failures without constant and exhaustive stress testing. This motivates approaches that preserve lossless generation quality while still exploiting lossy compression internally. FAFO fits into this class of methods, providing such reliability while delivering meaningful latency benefits.

## 1.2 KV CACHE COMPRESSION + N-GRAM CANDIDATE POOL DECODING BRINGS UNIQUE ADVANTAGES OVER TYPICAL SELF-SPECULATIVE DECODING

**Self-speculative decoding (SD) makes SD draftless and easier to deploy** While lossy KV cache compression methods might have their own pitfalls if employed in an end-to-end manner, it is common knowledge that not every word within a perfectly written sentence is hard to infer, as plain languages like English often contain many easy filler words that do not require outstanding model intelligence to predict correctly. In fact, this very idea fuels the established Speculative Decoding (SD) paradigm (Xia et al., 2022; Leviathan et al., 2023), where a small draft model with fewer resource demands first generates guesses, which are then verified with the larger target model. The main drawback of standard SD methods is that they involve non-trivial effort and resources to align and host a separate draft model (Yan et al., 2025; Li et al., 2024b;c; 2025b). A subfield named Self-Speculative Decoding (self-SD) has since been promoted by removing the need to host a separate draft model, where the draft tokens are instead generated by the same target model, but under a much

---

[2]See examples in Appendix C about the recorded failure modes of established methods like H2O (Zhang et al., 2023b), SnapKV (Li et al., 2024a), and more.

more resource-efficient mode. Specifically, methods like Sun et al. (2024); Sadhukhan et al. (2025); Zhang et al. (2023a); Xia et al. (2025) have explored the general idea of integrating lossy KV cache compression with self-speculative decoding, attempting to deliver lossless generation quality with improved latency.

**Self-SD's inherent shortcomings**  However, **we find all such self-speculative decoding methods come with at least one of the two practical shortcomings.** First, they all require maintaining separate sets of KV cache for the "draft" and "target" token generation, making them more memory-hungry than just doing full model inference. **This undercuts the very purpose of going draftless in the first place**: self-SD methods typically deliver worse latency performance than standard SD (as the latter can afford to craft and train an SD-specific draft model (Li et al., 2025b)), so the main goal of going draftless is to reduce memory demands, which is critical in resource-constrained scenarios like local hosting. Secondly, we find many such self-SD methods lack general usability: for instance, TriForce (Sun et al., 2024) demands a tiny long-context model that shares the target model's vocabulary, and can be as slow as $0.17\times$ the full model inference under certain scenarios. SS (Zhang et al., 2023a) can require 7-to-20+ hours of task-specific optimization before it is ready for inference, MagicDec (Sadhukhan et al., 2025) only performs well under large batch sizes, and, in practice, its implementation limits its maximum generation length to just 96 tokens or lower (Wu et al., 2025). While we respect all prior art for their contributions, we believe it is fair to argue that such sensitivity makes them less ready for real-world deployment.

**N-gram decoding: immediate benefits and integration challenges**  With this in mind, we look into other lossless efficient decoding channels and find the n-gram candidate pool paradigm pioneered by Lookahead Decoding (Fu et al., 2024) to be a potential candidate. **Different from the sequential draft-then-verify design of SD, n-gram decoding generates its newly drafted n-grams in parallel with its lossless verification** (in Lookahead, both are done with full KV cache). This allows n-gram decoding to achieve a single KV cache footprint and, by design, completely sidesteps the first shortcoming mentioned above. We also find Lookahead to be much more robust to task scenarios in comparison to methods like TriForce (though it still breaks under certain workloads)

However, n-gram decoding has its own quirks. Most significantly, since drafting and verification occur within the same forward pass, it requires non-trivial system engineering efforts to support any kind of KV cache compression method in a meaningful way. This is vastly different from standard SD, where the sequential pipeline allows trivial access to almost all KV cache compression methods, since one can simply engage the drafting forward passes with compression and the verification ones without. To the best of our knowledge, no prior work has successfully integrated n-gram decoding with lossy KV cache compression, resulting in some major bottlenecks — e.g., Lookahead is unable to host a large number of n-gram guesses, making its end-to-end latency advantage less significant than the n-gram potential would otherwise allow (Fu et al., 2024; Xia et al., 2025).

**FAFO's advantages and contributions**  To bridge the gap, we present the FAFO framework: where we utilize the model with compressed KV cache to Fumble Around with great freedom and efficiency, collecting n-gram guessed tokens and storing them in a candidate pool, meanwhile Find Out at the same forward pass with the full KV cache. We present FAFO as a general framework that is compatible with all typical static or dynamic token dropping-based KV cache compression methods as a means of generating guessed tokens. In summary, our main advantages and contributions are as follows:

- **Leveled-memory, lossless n-gram decoding.** FAFO operates under a leveled memory footprint (to full-model inference) while using a *single* KV cache and maintaining lossless quality. In contrast, lossy KV cache compression methods sacrifice quality, and (self) SD methods usually require enlarged or duplicated KV caches to stay lossless. These advantages are largely inherited from the n-gram paradigm and characterize FAFO's benefits over self-SD methods; to the best of our knowledge, only Lookahead Decoding shares this trifecta (Fu et al., 2024).
- **Customized KV cache manager that leverages FlexAttention.** We develop a custom KV cache manager that leverages FlexAttention (Dong et al., 2025) as an interface to connect the n-gram decoding paradigm and token dropping–based KV cache compression under our FAFO framework, enabling flexible exploration of future n-gram methods with a variety of lossy KV schemes — without the grind of custom kernel development.

- **General usability across scenarios.** We evaluate far more downstream tasks than most efficient decoding works and empirically show that FAFO is more robust than typical self-SD methods across models, tasks, input sequence lengths, etc., delivering consistent speedups with strong quality preservation. While Lookahead is already a decently robust method, FAFO enjoys a noticeable performance lead over Lookahead and remains performant under task scenarios where Lookahead is not (e.g., Table 3).
- **Revive a stagnant paradigm.** The FAFO framework opens up a new avenue of efficient draftless decoding leveraging KV cache compression techniques. We argue that this breakthrough is particularly significant, as there has not been another lossless n-gram decoding method since the initial Lookahead Decoding, which debuted in late 2023. This contrast is especially striking, since within the same timeframe, we have seen a cluster of lossy KV cache–based SD methods developed despite their innate shortcomings (Zhang et al., 2023a; Elhoushi et al., 2024; Liu et al., 2024; Xia et al., 2025; Sun et al., 2024; Sadhukhan et al., 2025), calling for a revisitation of n-gram decoding to practically materialize its preferable properties.

## 2 RELATED WORKS

**Self-Speculative Decoding**    To the best of our knowledge, four self-SD works have touched on SD idea under a strict draftless context: Self-Speculative (SS) (Zhang et al., 2023a), TriForce (Sun et al., 2024)[3], MagicDec (Sadhukhan et al., 2025), and SWIFT (Xia et al., 2025). Specifically, SS and SWIFT leverage layer-skipping as the means to efficiently generate draft tokens, whereas TriForce and MagicDec integrate with KV cache compression methods like SnapKV (Li et al., 2024a) and LM-Infinite/StreamingLLM (Han et al., 2024; Xiao et al., 2024).

**FAFO differs from SD methods by only requiring one model and one set of KV cache.** Compared to the four self-SD methods like TriForce and SWIFT, FAFO prevails in requiring just one set of KV cache. This is because SD methods operate in a sequential draft-then-verify way, where some of the newly generated tokens will always rely on a lossy KV cache. We provide a detailed walkthrough on why this drawback is innate to SD methods in Appendix D. Further, **FAFO is empirically much more scenario-agnostic and overall more performant than such self-SD methods** (see Table 2). Finally, technically speaking, FAFO also differs from SD in its parallel verification process, which we will discuss in the next paragraph.

**N-Gram Candidate Pool Decoding**    Among efficient yet lossless decoding pipelines, n-gram decoding presents a unique paradigm. Developed from the Jacobi Decoding process (Santilli et al., 2023) and first proposed under Lookahead Decoding (Fu et al., 2024), n-gram decoding generates multiple guessed tokens and stores them in an n-gram candidate pool. It can then generate additional n-gram candidates and verify them against existing ones under the same forward pass in a truly parallel manner. This stands in contrast to the sequential process of all SD methods, where guessed tokens must first be generated by the draft model and only then verified by the target. We note that, while some prior art often introduces n-gram decoding (and specifically Lookahead Decoding) under the same realm as speculative decoding, **the n-gram and speculative decoding paradigms are completely different, given the parallel draft-and-verify vs the sequential draft-then-verify distinction.** Unique opportunities and challenges arise across each pipeline, where careful considerations must be made. For instance, this parallel design grants n-gram decoding several unique properties. Most notably, it allows n-gram decoding to take advantage of guessed n-gram tokens that are correct further down the decoding path, but not immediately as the next decoded token — something SD cannot do (and therefore must invest decent effort to make sure the draft and the target are aligned (Li et al., 2025b; Yan et al., 2025)). This phenomenon is extremely common, as from a linguistic standpoint, some degree of repetition is often needed to form a cohesive paragraph. Such repetition is exemplified by their blog's first GIF (Figure 1), which we recommend readers check out to get an intuitive sense of how frequently this occurs.

FAFO's verification process follows this n-gram candidate pool design. However, different from Lookahead Decoding, **FAFO's guessed token generation employs a cache-compressed version of the target model, allowing it to generate many more n-gram candidates within the same forward pass.** We emphasize that although this update sounds simple, it involves non-trivial system

---

[3]TriForce is in fact not strictly draftless, see Appendix C for details.

optimization, as a naive attempt to integrate KV cache compression with n-gram decoding would fail: the expensive overhead of mask recomputation during each decoding step would cancel out any latency improvements. However, with our custom KV cache manager, such integration can be efficiently achieved, addressing key n-gram challenges like how to host a large number of n-gram guesses effectively (which bottlenecks Lookahead as demonstrated in Figure 1). Additional improvements over Lookahead Decoding are discussed in Section 4.

## 3 MOTIVATION: PRACTICAL OVERVIEW OF FAFO'S ADVANTAGES

As detailed in Section 2, KV cache compression and self-speculative decoding have already seen mature development, while n-gram decoding — though stagnant in terms of progress — is also an established paradigm, first introduced in late 2023 (Yuan et al., 2024; Xia et al., 2024; Fu et al., 2024). Thus, much of the contribution of our work lies in whether our proposed FAFO can present significant advantages over these existing solutions — which we shall gladly report that it can.

Given the highly technical nature of n-gram decoding (Fu et al., 2024) and the lack of any lossless follow-up to Lookahead (in terms of being draftless, doing parallel verification, and using an n-gram pool) until FAFO, we expect that many of our audience to be unfamiliar with the implementation details of Lookahead Decoding. Thus, for a smoother delivery, we first present FAFO at a high level, comparing it against existing designs under controlled settings; **readers can view this section as a practical overview of FAFO's advantages** before we introduce its design and implementation details in Section 4. It is our honest assessment that the original Lookahead Decoding manuscript does not clearly convey how these n-grams are generated and reused, but Figure 4 (a GIF) of its accompanying blog illustrates the mechanism far more clearly. **We strongly encourage readers to check out this Figure 4 GIF[4] before proceeding, as it provides the clearest walkthrough of Lookahead Decoding**, and understanding this algorithm is essential to understanding FAFO.

At a high level, FAFO can be viewed as a major improvement over Lookahead with two main components: Fumble Decoding and Find Out Verification. During decoding, Fumble Decoding allows FAFO to generate n-gram guesses using compressed KV cache, rather than the full cache required in Lookahead. In practice, this is enabled by leveraging compression methods such as LM-Infinite/StreamingLLM (Han et al., 2024; Xiao et al., 2024), SnapKV (Li et al., 2024a), Quest (Tang et al., 2024), and more under our FlexAttention-powered (Dong et al., 2025) custom KV cache manager (Appendix E). Find Out Verification further improves upon Lookahead by caching more than just the most recently decoded token. More details are provided in Section 4.3.

In the following sections, we will conduct investigatory pilot studies to compare FAFO with its comparative works (self-SD methods and Lookahead) and show why FAFO's design is advantageous in addressing the pain points that are rather inherent to these schools of work. Specifically, we show that FAFO is more memory efficient and task-agnostic than self-SD methods (Section 3.1), and capable of hosting more n-grams (and therefore much more performant) than Lookahead (Section 3.2).

### 3.1 FAFO PRESENTS SIGNIFICANT MEMORY SAVINGS OVER SELF-SPECULATIVE DECODING METHODS

Self-SD methods reduce memory footprint compared to standard SD by eliminating the need for a separate draft model. Naturally, self-SD methods like TriForce (Sun et al., 2024) and SWIFT (Xia et al., 2025) tend to leverage compressed KV cache for draft token generation. However, they still require hosting multiple sets of KV cache, as the draft forward must be computed independently from the target forward with full cache (see Appendix D for details).

Table 1 shows that TriForce occupies a much larger memory footprint than FAFO. We also observe that under challenging multi-turn tasks like MT-Bench (Zheng et al., 2023), TriForce is significantly slower than simply running inference on the full model. This is because it requires guessed tokens to be generated at the exact right time and position — a condition that is difficult to meet, as reflected by the $\tau = 1.06$ theoretical upper bound in Table 2. In contrast, FAFO with LM-Infinite/StreamingLLM (Han et al., 2024; Xiao et al., 2024) as the backbone KV cache compression method achieves a $1.91\times$ practical speedup with a $2.29\times$ theoretical upper bound — a significant improvement over TriForce.

---

[4]Figure 4 of `https://lmsys.org/blog/2023-11-21-lookahead-decoding/`.

Table 1: Llama-2-7b-chat over MT-Bench, with TriForce additionally utilizing a llama-68m as the draft-draft model, following its own configuration. $\tau$ is the average acceptance length, a theoretical upper bound of practical speedup if no overhead is considered.

| Method | Peak Memory (MB) | Speedup | $\tau$ |
|---|---|---|---|
| FAFO-Stream | **2362** | **1.91×** | 2.29 |
| Lookahead | **2183** | **1.61×** | 1.66 |
| FAFO-2forward | 3455 | 1.19× | 2.18 |
| TriForce | 4584 | 0.21× | 1.06 |
| SWIFT | 6196 | 1.17× | 2.65 |

To determine whether FAFO's gains over TriForce stem from the n-gram candidate pool allowing more flexible guessed token positions than speculative decoding, or from KV cache compression being more effective under parallel verification, we design an investigatory method called FAFO-2forward (Table 1). In FAFO-2forward, we drop the parallel verification design; instead, we perform one forward pass with compressed KV cache to collect n-gram guessed tokens, followed by a second forward pass to verify. Results show that FAFO-2forward is outperformed by the full FAFO-Stream by 60% (1.91× vs. 1.19×) in practical latency speedup, despite having similar theoretical average acceptance lengths ($\tau = 2.29$ vs. 2.18). This indicates that while maintaining an n-gram candidate pool does provide an inherent advantage over strict sequential speculative generation when it comes to integration with KV cache compression means, the pipeline is only fully leveraged when combined with parallel verification upon fumble-generated tokens — justifying the integrated framework proposed by FAFO.

## 3.2 FAFO Allows for Much More Guessed N-Gram Generation than Lookahead Decoding

One major distinction between FAFO and Lookahead Decoding is that FAFO's n-gram guessed tokens are generated using only compressed KV cache (Section 4.2). This design allows FAFO to generate far more guesses than Lookahead, overcoming a key limitation recognized by the Lookahead authors (see Figure 8 of (Fu et al., 2024), where Lookahead's practical speedup drops sharply as the number of guesses increases). As shown by the blue lines in Figure 1, FAFO can generate many more n-gram guesses without incurring a speed penalty, whereas Lookahead peaks around 10 guesses and then suffers decreasing throughput as the number of guesses increases. Note that the number of hosted n-gram guesses almost directly determines the overall performance of a candidate pool-based method, since more guesses provide more opportunities for matches.

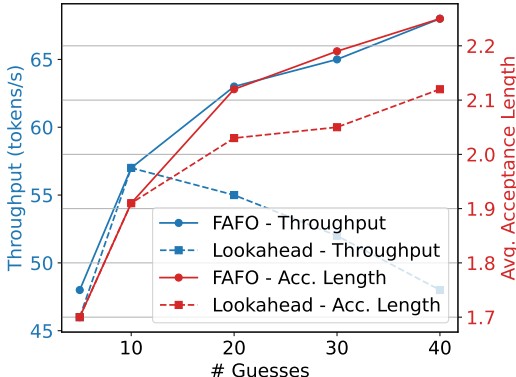

Figure 1: FAFO vs Lookahead Decoding with increased number of n-gram guesses. It can be observed that FAFO significantly outperforms in terms of both practical speedup and theoretical n-gram generation quality.

Beyond Fumble Decoding, we also introduce a small but critical improvement at the verification stage: Find Out Verification (Section 4.3). Instead of caching n-grams solely based on the most recently generated token, we assign a lookback window and cache them as part of the n-gram. Combined with Fumble Decoding, this enables FAFO to generate both more numerous and higher-quality n-grams, as reflected by the red lines in Figure 1, which show the theoretical upper bound of the average token acceptance length.

## 4 FAFO: Fumble Around and Find Out

FAFO implements a **unified execution pipeline** that decouples token generation logic from physical memory constraints. Specifically, the system orchestrates two concurrent workloads: (1) *Fumble Around*, which utilizes a compressed KV cache to generate high-throughput speculative guesses in parallel; and (2) *Find Out*, which verifies these previously generated guesses against the full set of

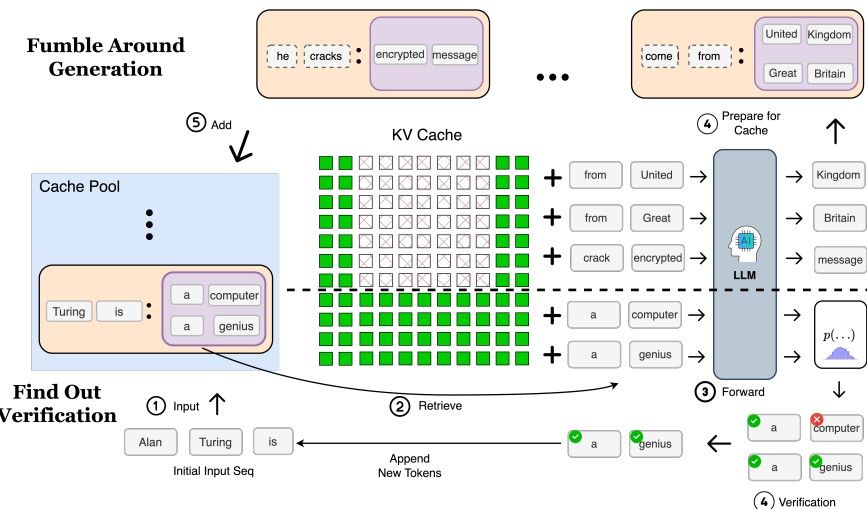

Figure 2: General Pipeline of FAFO. **Walkthrough of FAFO decoding** with (1) the input prompt *"Alan Turing is"*. (2) FAFO retrieves 2-gram candidate sequences such as *"a computer"* and *"a genius"* based on the suffix *"Turing is"*. (3) In parallel, Fumble Decoding generates next tokens of guesses like *"from United"*. A single forward pass is executed for both the Fumble Decoding and Find Out Verification branches. (4.1) Verified candidates (e.g., *"a genius"*) are accepted and appended to the input. (4.2) Newly generated 2-grams (e.g., *"United Kingdom"*) are cached by combining with previously buffered tokens from the same speculative subsequence (e.g., *"come from"*). (5) These new cached sequences are then added to the shared cache pool for reuse.

KV cache. To maximize GPU utilization, FAFO fuses these operations into a *single forward pass* via a custom sparse attention kernel (Section 4.4). This design amortizes the cost of loading QKV blocks over both drafting and verification, eliminating the synchronization overhead of separate draft/verify kernels typical in standard speculative decoding, while enabling multi-token acceptance per step (Figure 2).

## 4.1 PRELIMINARY

We consider a language model $p$, and a full sequence of tokens available at a given point in decoding $x_{1:|x|}$, consisting of both the initial prefill tokens and all tokens generated so far, where $|x|$ denotes the total number of tokens. Associated with $x_{1:|x|}$ is a set of key-value (KV) cache entries, denoted by $\mathrm{kv}_{1:|x|}$. Let $y^i_{s_i+1:s_i+k} = (y_{s_i+1}, y_{s_i+2}, \ldots, y_{s_i+k})$ denote a subsequence of $k$ tokens, where the index range $(s_i+1:s_i+k)$ refers to *absolute positional indices*, assuming the tokens in subsequence $i$ are placed at positions $s_i$ later in the current decoded sequence $x_{1:|x|}$. FAFO maintains a set of $n$ such subsequences as speculative future subsequences. We also define a KV cache compression function $\mathcal{C}$, i.e. methods such as LM-StreamingLLM (Xiao et al., 2024).

## 4.2 FUMBLE DECODING

FAFO uses a compressed set of KV cache to speculate future tokens. At each decoding step, FAFO maintains a set of $n$ independent speculative subsequences $y^1_{s_1+1:s_1+k}, y^2_{s_2+1:s_2+k}, \ldots, y^n_{s_n+1:s_n+k}$. Given the current KV cache entries $\mathrm{kv}_{1:|x|}$ and a KV cache compression function $\mathcal{C}$, FAFO leverages the compressed cache $\mathcal{C}(\mathrm{kv}_{1:|x|})$ to generate the next token for each speculative subsequence $y^i$ in parallel with negligible memory bandwidth cost as:

$$y^i_{s_i+k+1} = \arg\max p(y^i_{s_i+k+1}|y^i_{s_i+1:s_i+k}, \mathcal{C}\left(\mathrm{kv}_{1:|x|}\right)) \, \forall i \in \{1, \ldots, n\} \tag{1}$$

To ensure continuous generation of future tokens, the oldest token in each subsequence, namely $y^i_{s_i+1}$, is discarded after every decoding step. The remaining subsequence $y^i_{s_i+2:s_i+k+1}$ is then collected and offloaded to a CPU-side pool for later use in the "Find Out" verification phase with FAFO's prefix caching strategy.

**FAFO's prefix caching strategy.** For each guess stream $i$, FAFO maintains a buffer of the last $k$ discarded tokens, $y^i_{s_i-k+2:s_i+1}$, and uses this buffer to organize how newly generated subsequences are cached. Each new guess subsequence $y^i_{s_i+2:s_i+k+1}$ is cached under multiple discarded-token prefixes of increasing length up to $k$ (Algorithm 2):

$$(y^i_{s_i+1}), \quad (y^i_{s_i}, y^i_{s_i+1}), \quad \ldots, \quad (y^i_{s_i-k+2:s_i+1}).$$

Later, during verification, when the current decoding context ends with a particular prefix, the system retrieves all cached subsequences associated with that prefix and reuses them as candidate guesses. Because intuitively longer matching prefixes share more tokens with the current context, the corresponding candidates have a higher probability of matching the model's distribution. Proposition F.1 theoretically bounds the effectiveness of this strategy: longer matching prefixes correlate with lower perplexity, prioritizing candidates with the highest probability of acceptance. Thus, the verification phase prioritizes candidates indexed under longer prefixes, improving the quality of verified guesses while keeping the guess pool entirely on the CPU and avoiding additional GPU memory overhead.

### 4.3 Find Out Verification

FAFO retrieves $n$ guesses (speculative subsequences) $a^1, \ldots, a^n$, previously generated by Fumble Around decoding, from the CPU-side cache pool based on the current decoded prefix (Alg. 3; Fig. 2). Concretely, FAFO considers up to $k$ of the most recent tokens in the current decoded sequence, $x_{|x|-m+1:|x|}$ for $1 \leq m \leq k$, and uses these length-$m$ suffixes as lookup keys into the cache, starting from the longest $m$ for which entries exist until we have retrieved enough $n$ guesses. Each cached guess was originally indexed under the corresponding discarded-token prefix when it was generated, so this lookup returns candidates whose prefix context matches the current decoding state as closely as possible.

The retrieved candidates are then verified in parallel using the full KV cache via distribution matching, following the standard speculative- decoding accept/reject rule of Leviathan et al. (2023): the target LLM is forwarded on the draft tokens, and each token is accepted if and only if the model's predicted next token exactly matches the draft token (Alg. 4). Unlike Lookahead, which caches and retrieves candidates conditioned solely on $x_{|x|}$, FAFO conditions retrieval on a longer suffix, yielding higher-quality guesses. After verification, all accepted tokens are appended at once before the next iteration, and in practice more than one token is typically accepted, translating into lower end-to-end decoding latency.

### 4.4 System-Efficient FAFO via Sparse Attention Kernels

FAFO implements a *draftless* pipeline by fusing the Fumble Around (speculation) and Find Out (verification) phases into a single model forward pass. By concatenating draft tokens and verification tokens, we leverage a unified attention mask (Figure 4) to maximize GPU utilization. However, simply masking out tokens logically does not guarantee system efficiency.

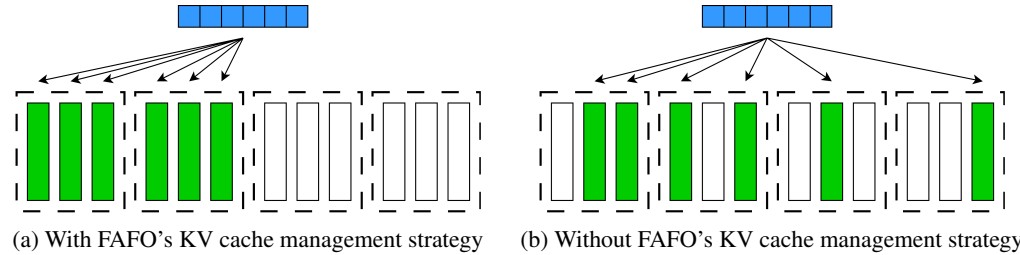

(a) With FAFO's KV cache management strategy      (b) Without FAFO's KV cache management strategy

Figure 3: **Logical Sparsity vs. Physical Fragmentation.** (a) **With FAFO:** By actively compacting relevant tokens (green) into a dense region, FAFO ensures that logical sparsity translates to reduced physical I/O. The kernel iterates only over the active block budget $\mathbf{B}_{fixed}$, effectively terminating the memory load early. (b) **Without FAFO:** Relevant tokens are fragmented across the address space. Because FlexAttention loads data at **block granularity**, the hardware performs redundant HBM transactions on blocks containing mostly irrelevant (white) data, limiting bandwidth utilization.

**The Memory Wall in Sparse Attention.** While the logical attention matrix for FAFO is highly sparse, standard attention kernels are agnostic to this structure. They typically load the entire KV cache from High Bandwidth Memory (HBM) into on-chip SRAM to compute attention scores, applying masks only *after* the expensive memory load operations. Consequently, the decoding latency remains **memory-bandwidth bound**, scaling linearly with the full context length $|x|$ rather than the sparse subset size. To overcome this, we employ FlexAttention to enforce block-sparse computation, but its naive application introduces new overheads.

**The Challenge of Dynamic Masking.** While FlexAttention is well-suited for the prefill phase, where the input length remains static, applying it to the decoding phase is highly non-trivial. The context grows with each generated token, requiring the block mask to be recomputed at every step. **This recomputation is prohibitively costly, rendering naive use of FlexAttention during decoding impractical** without a decoding-aware block masking scheme specifically designed to handle dynamic attention contexts. Specifically, dynamic mask regeneration triggers *repeated graph capture and compilation overheads*, effectively negating the computational benefits of sparsity by stalling the GPU pipeline.

**Hardware-Aware KV Layout for Zero-Overhead Masking.** We resolve this by decoupling the *logical* position of tokens from their *physical* memory location. We implement a **Physically Contiguous, Logically Sparse** memory management strategy. We allocate a fixed buffer of physical KV blocks, $\mathbf{B}_{fixed}$, at the head of the KV tensor to store the compressed entries $\mathcal{C}(\mathrm{kv}_{1:|x|})$.

$$\text{KV Cache Physical Layout} = \left[ \underbrace{\mathcal{C}(\mathrm{kv}_{1:|x|})}_{\text{Selected Blocks}} \; \| \; \underbrace{\mathrm{R}(\mathrm{kv}_{1:|x|})}_{\text{Irrelevant Blocks}} \right] \tag{2}$$

By enforcing that all relevant tokens for the current "Fumble Around" step reside within $\mathbf{B}_{fixed}$, we construct a static FlexAttention block mask that instructs the kernel to load *only* the first $|\mathbf{B}_{fixed}|$ blocks from HBM to SRAM. Irrelevant entries $\mathrm{R}(\mathrm{kv})$ are logically preserved but physically skipped during the attention kernel execution (Algorithm 1).

**Inference KV Cache Management.** Between decoding steps, we execute an asynchronous device-to-device copy to swap newly selected tokens by function $\mathcal{C}$ into $\mathbf{B}_{fixed}$ and evict expired tokens to the tail (Appendix H). This guarantees that the FlexAttention kernel operates on a dense, contiguous memory region, shifting the workload from an IO-bound to a compute-bound regime. In contrast, the vanilla implementation suffers from severe block-level fragmentation, where valid KV entries are sparsely distributed across the physical address space. Because FlexAttention loads data at the granularity of blocks, this fragmentation forces the kernel to incur redundant HBM transactions, fetching entire physical blocks even when they contain negligible valid context (Figure 3). By enforcing a defragmented, dense memory layout that eliminates the overhead of dynamic mask regeneration, FAFO maximizes effective memory bandwidth utilization, directly translating logical sparsity into realized wall-clock acceleration.

## 5 EXPERIMENTS

**Models and Settings** Our core evaluations focus on Llama-2-Chat (7B) (Touvron et al., 2023), LLaMA-3-Instruct (8B) (Grattafiori et al., 2024), LLaMA-3.1-Instruct (8B) (Grattafiori et al., 2024), and Qwen2.5-Instruct (7B, 32B) (Yang et al., 2024), providing fair coverage of different attention architectures, model families, and scales. Additionally, we evaluate DeepSeek-R1-Distill (Qwen-7B and Llama-8B) (Guo et al., 2025) under reasoning tasks. All experiments are conducted on a single NVIDIA A100 GPU with 80GB of memory. Unless otherwise specified, all models are served with FP16 precision and a batch size of 1, following the setup of existing latency-oriented works (Cai et al., 2024; Fu et al., 2024). FlexAttention kernels (Dong et al., 2025) with our customized KV cache manager are used for efficient sparse attention computation.

**Benchmarks and Metrics** Following prior speculative decoding works (Fu et al., 2024; Li et al., 2024b), we evaluate FAFO on three widely used benchmarks: MT-Bench (Zheng et al., 2023), GSM8K (Cobbe et al., 2021), and HumanEval (Chen et al., 2021). Additionally, we test FAFO on Multi-IF (He et al., 2024), SCBench (Li et al., 2025a), LongBench (Bai et al., 2023), and AIME 24 to further demonstrate performance under multi-turn, long-context, and long-generation scenarios.

Finally, we include PG19 for alignment with TriForce. Following Fu et al. (2024); Li et al. (2024b); Sadhukhan et al. (2025), we report FAFO's performance with the following metrics:

- **Wall-clock speedup ratio**: The observed speedup relative to vanilla autoregressive decoding, measured in tokens per second.
- **Average acceptance length** $\tau$: The average number of tokens accepted per decoding step.

**Baselines**   We compare FAFO against established self-SD and n-gram decoding methods, including TriForce (Sun et al., 2024), SWIFT (Xia et al., 2025), and Lookahead Decoding (Fu et al., 2024). We omit MagicDec (Sadhukhan et al., 2025), as its implementation cannot support more than 96 newly decoded tokens, making it incompatible with most challenging tasks. Older methods like SS (Zhang et al., 2023a) are also omitted, as they have been extensively compared with our featured baselines and no longer represent the SOTA baselines (Xia et al., 2025). "FAFO-Stream" and "FAFO-Quest" refer to our proposed FAFO method instantiated with LM-Infinite/StreamingLLM (Han et al., 2024; Xiao et al., 2024) and Quest (Tang et al., 2024) as the KV cache eviction methods. We chose these two because they are established representatives of static and dynamic token-dropping approaches. **While we could have developed a custom KV cache compression method to replace them**, **we intentionally avoid doing so** to respect prior art and to prevent reinventing the wheel under unnecessary conditions. More importantly, we believe the community benefits most from a general framework where they can experiment with different KV cache compression methods, rather than a baked implementation supporting only our own.

Table 2: Observed practical latency speedup and average acceptance length $\tau$ on MT-bench, GSM8K, and HumanEval-Completion.

| Models | Method | MT-bench | | GSM8K | | HumanEval-C | |
|---|---|---|---|---|---|---|---|
| | | Speedup | $\tau$ | Speedup | $\tau$ | Speedup | $\tau$ |
| **Llama-2-7b-chat** | FAFO-Stream | **1.91**× | 2.29 | **1.63**× | 2.70 | **2.03**× | 2.34 |
| | FAFO-Quest | 1.32× | 2.20 | 1.40× | 2.60 | 1.63× | 2.33 |
| | Lookahead | 1.61× | 1.66 | 1.58× | 1.65 | 1.72× | 1.77 |
| | TriForce | 0.21× | 1.06 | 0.17× | 1.12 | 0.22× | 1.06 |
| | SWIFT | 1.17× | 2.65 | 1.22× | 2.43 | 1.13× | 3.79 |
| **Llama-3-8B-Instruct** | FAFO-Stream | **1.58**× | 2.12 | **1.60**× | 2.10 | **1.65**× | 2.00 |
| | FAFO-Quest | 1.50× | 2.05 | 1.45× | 2.05 | 1.43× | 2.00 |
| | Lookahead | 1.49× | 2.01 | 1.39× | 1.99 | 1.52× | 1.90 |
| | SWIFT | 1.18× | 3.33 | 1.34× | 3.73 | 1.22× | 3.90 |
| **Llama-3.1-8B-Instruct** | FAFO-Stream | **1.50**× | 2.10 | **1.31**× | 2.08 | **1.57**× | 2.20 |
| | FAFO-Quest | 1.40× | 2.08 | 1.28× | 2.08 | 1.48× | 2.19 |
| | Lookahead | 1.21× | 2.01 | 1.25× | 2.01 | 1.28× | 2.10 |
| | SWIFT | 0.94× | 2.93 | 0.98× | 3.22 | 1.03× | 2.38 |
| **Qwen2.5-7B-Instruct** | FAFO-Stream | 1.43× | 2.10 | **1.60**× | 2.30 | **1.44**× | 2.10 |
| | FAFO-Quest | **1.46**× | 2.20 | 1.44× | 2.20 | 1.38× | 2.08 |
| **Qwen2.5-32B-Instruct** | FAFO-Stream | **1.20**× | 2.07 | 1.30× | 2.36 | **1.40**× | 2.33 |
| | Lookahead | 1.09× | 2.15 | **1.44**× | 2.42 | 1.26× | 2.36 |

**End-to-End Effectiveness**   Table 2 reports the wall-clock speedup ratio and average acceptance length of FAFO compared to other baselines. FAFO-Stream or FAFO-Quest essentially achieves the practical speedup and theoretical $\tau$ under all featured settings, approximately 30% faster than Lookahead Decoding. Specifically, we find FAFO to be robust under challenging tasks like MT-Bench and GSM8K — tasks that pose a significant challenge to methods like TriForce and SWIFT, which sometimes experience negative speedups.

Table 3: Speedup ratio and average acceptance length $\tau$ on datasets from LongBench.

| Models | Method | Multi-News | | LCC | | TREC | | Qasper | | 2WikiMQA | |
|---|---|---|---|---|---|---|---|---|---|---|---|
| | | Speedup | $\tau$ | Speedup | $\tau$ | Speedup | $\tau$ | Speedup | $\tau$ | Speedup | $\tau$ |
| **L3.1 8B** | FAFO-Stream | **1.97**× | 2.81 | **1.78**× | 3.08 | **2.01**× | 4.27 | **2.20**× | 3.54 | **1.94**× | 3.73 |
| | FAFO-Quest | 1.62× | 2.83 | 1.70× | 3.10 | 1.80× | 4.37 | 2.12× | 3.65 | 1.87× | 3.95 |
| | Lookahead | 1.12× | 1.93 | 1.48× | 2.58 | 1.54× | 3.69 | 0.85× | 2.19 | 1.01× | 2.94 |
| | SWIFT | 1.09× | 3.15 | 1.08× | 3.72 | 1.09× | 4.17 | 1.41× | 4.10 | 1.13× | 3.25 |

Given KV cache is most significant under a long context setting, we further feature some common tasks from LongBench to confirm FAFO's task robustness. According to Table 3, we are glad to report that FAFO tends to offer even better performance on long context tasks. This result is intuitive, as constant budget KV cache compression methods like LM-Infinite/StreamingLLM often offer the most efficiency gains under such long context settings.

## ETHICS STATEMENT

Given the technical focus of this work on algorithmic improvements for decoding efficiency via compressed KV cache with integrated verification, we do not identify specific limitations that require emphasis within the scope of our methodology. The design and evaluation of our approach are grounded in controlled latency benchmarking and lossless generation quality, and the method demonstrates consistent speedup without increasing memory overhead. While integration into production pipelines may require careful engineering, the underlying mechanism remains general and compatible with a wide range of KV cache compression strategies. Broadly, our work contributes to the ongoing effort to make large language models more efficient and deployable, potentially lowering the environmental and economic costs of inference. However, it also highlights the need for continued research into the failure modes of lossy compression methods and the development of robust safeguards for their deployment in high-stakes settings.

## REPRODUCIBILITY STATEMENT

We facilitate reproducibility through multiple artifacts and detailed documentation. A repository will be released in the future host source code, experiment scripts, and configuration files. Comprehensive procedural details are provided in Section 4, Section 5, Appendix E, Appendix F, and Appendix H, enabling independent verification and extension.

At present, public release of the code is temporarily deferred due to our employer's policies governing dissemination of software and research artifacts (including approvals related to IP ownership, confidentiality, and third-party licensing). We are in the process of obtaining the necessary authorizations and credentials for open-source release. Upon paper's publication, we will promptly publish the repository under an appropriate license.

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

## A    USAGE OF LARGE LANGUAGE MODELS

We would like to disclose that part of the writing of this paper was polished by a language model, though a human researcher is there to verify that the final output is true to the researcher's opinion.

## B    LIMITATIONS

One major limitation of FAFO — or n-gram decoding methods in general — is the lack of batch inference support, where all experiments are done under the most extreme latency-sensitive setup (batchsize = 1). While we do believe that draftless methods, by nature, make the most sense in local deployment or resource-constrained scenarios where batching is not a top priority, we also recognize that having such support would be a clear advantage.

We benchmarked the per-step FLOPs of both FAFO and Lookahead Decoding (Fu et al., 2024) and observed that n-gram methods typically incur substantially higher FLOPs than vanilla decoding. In contrast, speculative decoding (SD) methods, given their sequential draft-then-verify design, often exhibit lower average FLOPs than the original model (except under extreme draft model configurations). The high FLOPs consumption of n-gram methods is intuitive: instead of relying on a smaller draft model (as in SD) or performing a single full model forward pass (as in vanilla decoding), n-gram decoding performs full model verification *and* generates multiple new n-gram candidates within the same forward pass. This implies that, in a batched setting, n-gram methods are likely to hit the compute-bound regime much sooner, leaving less room for meaningful speedup.

For disambiguation, we are not claiming that batched n-gram decoding is impossible. In fact, we believe there are many promising avenues for improving the compute and memory efficiency of the n-gram pipeline established by Lookahead (e.g., one of them is to avoid concatenating n-gram guesses and use an "inner batch" mechanism with shared prefix cache to verify multiple guesses). Our point is simply that **developing a truly batch-capable n-gram method would require substantial new systems design, likely worthy of a dedicated follow-up paper.** We provide this section here to be transparent with our readers about this current limitation shared by FAFO, Lookahead, and the general n-gram paradigm. The readers are advised to carefully gauge whether the intended task scenario fits the n-gram paradigm's advantages before adopting our method.

## C    EXTENDED RELATED WORKS

**Token Dropping-based KV Cache Compression**    Due to the growing nature of the KV cache, many lossy compression techniques have been developed to reduce memory footprint and improve generation latency. For instance, LM-Infinite/StreamingLLM (Han et al., 2024; Xiao et al., 2024) preserves only the first few "attention sink" and recent tokens while dropping intermediate ones, achieving a constant KV cache budget. H2O (Zhang et al., 2023b) and SnapKV (Li et al., 2024a) evict tokens based on attention scores, representing static KV cache eviction methods. Dynamic counterparts like Quest (Tang et al., 2024) and NSA (Yuan et al., 2025) select retained tokens at each decoding step, unlike static methods which typically evict in one shot after prefill. **FAFO differs from these methods by offering lossless generation quality.** To our knowledge, nearly all KV cache compression techniques are lossy, with their failure patterns occasionally revealed via benchmarks like `longctx_bench` (Yuan et al., 2024) and SCBench (Li et al., 2025a). FAFO is well-suited for cases demanding both latency speedup and lossless outputs.

**Speculative Decoding**    Speculative Decoding (SD) uses a smaller draft model to generate guessed tokens and a larger target model to verify them, enabling the potential of confirming multiple tokens per single forward pass (Xia et al., 2022; Leviathan et al., 2023). Later works like SpecInfer (Miao et al., 2023) and Sequoia (Chen et al., 2024) use tree attention for efficient multi-token verification.

As mentioned in the Section 1, the main criticism of SD is the need to craft and host a separate draft model, which induces significant alignment efforts and resource demands — a major challenge for `r/LocalLLaMA`-like local hosting users. Thus, scholars have explored the potential of draftless SD, often known as *Self-Speculative Decoding* (self-SD), which adopts the same model as both draft and target, often with the draft forward being a sparse variant of the target forward.

**Strictly Draftless Methods** By "strict draftless context" in Section 2, we mean that there is one original model, with no parameter or architectural modification, serving as both draft and target. We clarify that TriForce (Sun et al., 2024) does not fit this description, as it is a three-level method where a tiny draft-draft model sharing the target model's vocabulary is used to conduct the first drafting, which is then processed by the target model with partial cache, and finally verified with the target model in full cache. Strictly speaking, TriForce is therefore not fully draftless. However, we include it here and use it as a major baseline because: a) if we remove the tiny draft-draft model, it is strictly draftless in the second and third stages, meaning that this additional stage is likely of significant value despite its small additional footprint; and b) TriForce was developed by the same lab as MagicDec (Sadhukhan et al., 2025), but is more performant under latency-sensitive (batchsize=1) scenarios, making it a major landmark to benchmark against FAFO. Hereinafter, we might refer to TriForce as a draftless/self-SD method for concise delivery.

**More Tangentially-Related Speculative Decoding Methods.** We note that there are additional SD methods that also leverage lossy compression, such as Kangaroo (Liu et al., 2024) and Layer-Skip (Elhoushi et al., 2024). However, Kangaroo requires training additional components on top of the original model, and LayerSkip demands weight updates. As a result, they either diverge from the "strictest draftless context," or are no longer lossless to the original model. These works are technically unrelated to FAFO by large, but we opt to feature them here because the distinction relies on an intricate understanding of such methods.

Further, we have plenty of SD, self-SD, or SD-adjacent methods that focus on long context performance. Other than the above-discussed TriForce (Sun et al., 2024) and MagicDec (Sadhukhan et al., 2025), works like LongSpec (Yang et al., 2025), QuantSpec (Tiwari et al., 2025), and TokenSwift (Wu et al., 2025) also contribute. Their connection with FAFO mostly resides in the fact that FAFO is also evaluated on many long context tasks.

**Failure Modes of Lossy KV Cache Compression Methods** Yuan et al. (2024) reveals that H2O (Zhang et al., 2023b) can perform decently on Needle-in-a-Haystack/passkey retrieval-like tasks (Mohtashami and Jaggi, 2023) if given a shorter passkey to retrieve and a continued prompt, but would fail catastrophically (dropping from 100% to 35%) once the passkey length is extended. Similarly, benchmarks like SCBench (Li et al., 2025a) and later works like Ada-KV (Feng et al., 2024) reveal that while strong token dropping methods like SnapKV (Li et al., 2024a) are often performant across many tasks, they face a significant performance degradation if they are unaware of the user query before eviction, thus directly hurting their multi-turn performance — arguably one of the signature capabilities of instruction-following LLMs. While this paragraph is in no way an exhaustive list of how lossy KV cache compressions would fail, it illustrates a recurring pattern that is worth attention: lossy KV cache compression introduces brittle, task-dependent weaknesses that are difficult to anticipate or detect without extensive, highly specific stress testing.

## D  TECHNICAL COMPARISON OF FAFO VS. SPECULATIVE DECODING METHODS

We first provide details explanation on why FAFO can achieve leveled memory footprint, while other SD baselines need to maintain addition KV cache. Recall that Speculative Decoding (SD) methods follow the sequential draft-THEN-verify pipeline. So for such methods to be effective, the draft model must generate multiple (and consecutive) draft tokens; THEN, such draft tokens are verified.

Let us take the simplified TriForce (Sun et al., 2024) (ignoring the 68M tiny draft-draft model) as an example. From a fresh start (given a certain length of input, with no output yet), we have the following steps:

1. **Initial Prefill:** TriForce first identifies a set of token chunks from the prefill as "important" and evicts the rest, forming a lossy cache. At this point, this lossy cache is still a strict subset of the full/exact cache.

2. **Draft Token Generation:** TriForce then sequentially decodes multiple draft tokens, where the lossy cache naturally grows. Since the newly decoded draft tokens are generated upon

the lossy KV, their own KV are also lossy. Thus, during draft generation, the lossy cache is no longer a subset of the full cache.

3. **Full Cache Verification & Lossy Cache Update:** After obtaining the draft tokens, Tri-Force engages in verification and obtains the full and exact cache of all accepted tokens. TriForce updates the lossy cache by replacing the accepted tokens' KV with the exact ones. It then evicts a number of "least important" tokens from the updated lossy cache to prevent its size from growing out of control.

4. **Repeat and Rebuild:** Steps 2 & 3 are repeated, and occasionally a full rebuild of the lossy cache is triggered, pending on various factors (e.g., low acceptance rate). TriForce cannot afford to materialize only one set of KV cache because of two main reasons: a) During Step 2, its lossy and exact cache copies diverge, where some lossy KV must be stored; and b) Although the lossy cache becomes a subset of the full cache again by the end of Step 3 verification, generating draft tokens upon this lossy cache would require an updated and relatively fine-grained slicing/gathering (compared to something like a StreamingLLM-style masking) upon the full cache — which is a fairly costly operation to engage in per each verification step.

We note that this stands in contrast to n-gram candidate methods like FAFO, where verification occurs in parallel with the guess token generation. At each step, exactly one token is added per each "input" (# of input = # of n-grams + # verification + 1). In this setting, the lossy and full caches never diverge, so storing just one copy of the KV cache is sufficient. However, realizing efficiency gains under this setting still requires non-trivial efforts, as a naïve attention mask provides negligible efficiency improvements over full attention and becomes a bottleneck in the parallel pipeline. To address this, we construct a static block-wise mask with FlexAttention and utilize our proposed swapping-based KV cache management design to maintain end-to-end efficiency.

# E  FAFO'S TOKENS STRUCTURE AND ATTENTION MASK

FAFO has both Fumble Around decoding and Find Out verification in a single forward pass of a single model, achieving a *draftless* setting: FAFO concatenates tokens from both the fumble decoding and verification phases, using a designated attention mask (Figure 4). In Fumble Around, instead of organizing the $n$ subsequences in a row-wise manner, we organize them in a column-wise manner (i.e., grouping tokens at the same position across subsequences step-by-step: $y_1^1, y_1^2, \ldots, y_1^n, y_2^1, y_2^2, \ldots, y_2^n, \ldots$), following the practice in Fu et al. (2024). The attention mask of tokens in fumble around decoding part is then constructed such that each token can only attend to tokens that (i) appear earlier in the sequence (i.e., have smaller position indices) and (ii) belong to the same subsequence (i.e., share the same column index). This column-wise organization significantly simplifies updating the sequence when new tokens are generated. Instead of shifting existing tokens and inserting new ones (as would be needed in a row-wise layout), FAFO only needs to *append* the newly generated tokens $y_{k+1}^1, y_{k+1}^2, \ldots, y_{k+1}^n$ to the end of the concatenated sequence. In contrast, tokens in the verification part follow the standard causal attention masking.

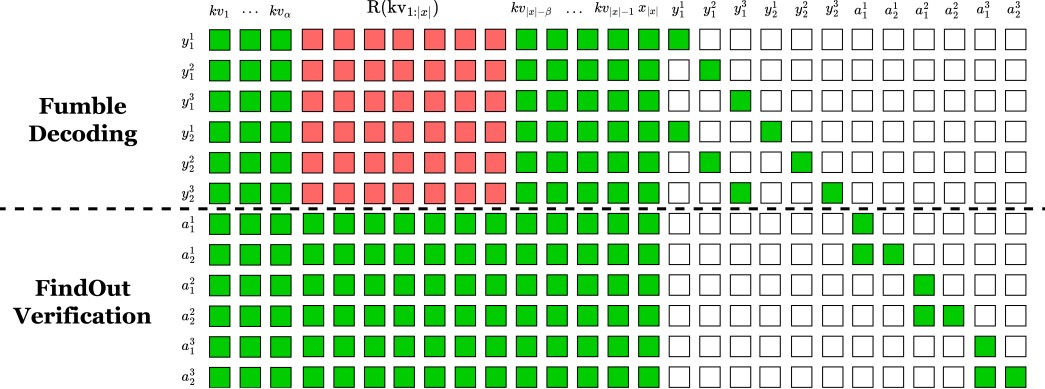

Figure 4: Attention mask for FAFO's concatenation of fumble around decoding and find out verification. Here, $x_{|x|}$ represents the latest token in the sequence while $kv_1 \ldots kv_{|x|-1}$ are the KV cache of previous tokens. Tokens $a_1^1, \ldots, a_2^2$ correspond to the verification phase and are flattened into a 1-D tensor with standard causal masking. Tokens $y_1^1, \ldots, y_2^2$ correspond to the fumble decoding phase and are flattened into a 1-D tensor using the designated masking described in Section 4.4 and Appendix E.

# F  CACHE POOL MANAGEMENT

Although cached subsequences are offloaded to the CPU pool to save GPU memory, the pool size is expected to grow linearly, adding more load to the verification process later. On the other hand, subsequences generated earlier have a lower chance of being accepted later as the decoded sequence length increases more and more. To mitigate this, we limit the number of cached subsequences per starting token to $n$. Older candidate subsequences are evicted in a least-recently-used (LRU) manner.

**Proposition F.1** (Monotonicity of verification success under longer matching prefixes). *Let $p_\theta$ be an autoregressive language model over sequences $y_{1:T}$,*

$$p_\theta(y_{1:T}) = \prod_{t=1}^{T} p_\theta(y_t \mid y_{1:t-1}).$$

*Fix a ground-truth sequence $y_{1:T}^\star$ and define, for any prefix length $\ell < T$,*

$$q_\ell := p_\theta(y_{\ell+1:T}^\star \mid y_{1:\ell}^\star) = p_\theta(y_{\ell+1}^\star, \ldots, y_T^\star \mid y_1^\star, \ldots, y_\ell^\star).$$

*Then for all $\ell < T-1$,*

$$q_{\ell+1} \geq q_\ell.$$

*In other words, the probability that the model exactly reproduces the remaining suffix of $y^\star$ is non-decreasing in the length of the matching prefix.*

*Proof.* By the chain rule for $p_\theta$ and the definition of $q_\ell$,

$$q_\ell = p_\theta\left(y^\star_{\ell+1:T} \mid y^\star_{1:\ell}\right) = p_\theta\left(y^\star_{\ell+1} \mid y^\star_{1:\ell}\right) p_\theta\left(y^\star_{\ell+2:T} \mid y^\star_{1:\ell+1}\right).$$

The second factor on the right-hand side is exactly $q_{\ell+1}$:

$$q_{\ell+1} = p_\theta\left(y^\star_{\ell+2:T} \mid y^\star_{1:\ell+1}\right).$$

Hence

$$q_\ell = p_\theta\left(y^\star_{\ell+1} \mid y^\star_{1:\ell}\right) q_{\ell+1}.$$

Since $p_\theta\left(y^\star_{\ell+1} \mid y^\star_{1:\ell}\right) \in (0, 1]$, we obtain

$$q_{\ell+1} = \frac{q_\ell}{p_\theta\left(y^\star_{\ell+1} \mid y^\star_{1:\ell}\right)} \geq q_\ell.$$

Thus $q_{\ell+1} \geq q_\ell$ for all $\ell < T - 1$, which proves the claim. $\qquad\square$

# G    Additional Experiments

Due to page limitations, we can only include so many experiments in the main text. Here, we present additional experiment results on FAFO under a non-greedy decoding setting, its performance under reasoning task (as with reasoning models tend to decode a lot of chain-of-thought tokens before delivering the final answer, making up a huge presence in KV cache), and how it holds up again SCBench[5], which is often regarded as one of the hardest benchmark for lossy KV cache compression methods. Last, we conduct ablation studies on hyper-parameters crucial to FAFO.

## G.1    Sampling Decoding

Table 4 presents FAFO vs Lookahead under non-greedy sampling setting. For each model, we adopt their default `generation_config.json` for sampling setting as different model might prefer a different sampling setup. Specifically, we have:

- **Llama-2-7b-chat**: temperature $T = 0.6$, top-p $= 0.9$
- **Llama-3-8B-Instruct**: temperature $T = 0.6$, top-p $= 0.9$
- **Llama-3.1-8B-Instruct**: temperature $T = 0.6$, top-p $= 0.9$
- **Qwen2.5-7B-Instruct**: temperature $T = 0.7$, top-p $= 0.8$, top-k $= 20$
- **Qwen2.5-32B-Instruct**: temperature $T = 0.7$, top-p $= 0.8$, top-k $= 20$

Table 4: Observed practical latency speedup and average acceptance length $\tau$ on MT-bench, GSM8K, and HumanEval-Completion with sampling temperature $T > 0$.

| Models | Method | MT-bench | | GSM8K | | HumanEval-C | |
|---|---|---|---|---|---|---|---|
| | | Speedup | $\tau$ | Speedup | $\tau$ | Speedup | $\tau$ |
| **Llama-2-7b-chat** | FAFO-Stream | **1.91**× | **2.25** | **1.95**× | **2.72** | **1.87**× | **2.35** |
| | FAFO-Quest | 1.78× | 2.19 | 1.71× | 2.60 | 1.87 × | 2.34 |
| | Lookahead | 1.64× | 2.01 | 1.68× | 2.50 | 1.79× | 2.33 |
| **Llama-3-8B-Instruct** | FAFO-Stream | **1.58**× | **2.06** | **1.57**× | **2.09** | **1.58**× | **2.00** |
| | FAFO-Quest | 1.47× | 2.00 | 1.46× | 2.02 | 1.49× | 2.01 |
| | Lookahead | 1.40× | 1.99 | 1.43× | 2.03 | 1.50× | 1.98 |
| **Llama-3.1-8B-Instruct** | FAFO-Stream | **1.39**× | 2.08 | **1.41**× | **2.10** | **1.58**× | **2.22** |
| | FAFO-Quest | 1.37× | **2.10** | 1.36× | 2.07 | 1.50× | 2.20 |
| | Lookahead | 1.28× | 2.00 | 1.28× | 2.03 | 1.45× | 2.08 |
| **Qwen2.5-7B-Instruct** | FAFO-Stream | **1.48**× | **2.16** | **1.56**× | **2.30** | **1.55 ×** | **2.10** |
| | FAFO-Quest | 1.41× | 2.03 | 1.48× | 2.26 | 1.41× | 2.07 |
| **Qwen2.5-32B-Instruct** | FAFO-Stream | **1.25**× | 2.10 | **1.35**× | **2.40** | **1.44**× | **2.35** |
| | FAFO-Quest | 1.12× | 2.02 | 1.23× | 2.32 | 1.33× | 2.35 |
| | Lookahead | 1.20× | **2.12** | 1.16× | 2.35 | 1.27× | 2.30 |

## G.2    Robustness under Reasoning-Intensive Task

Table 5: Speedup ratio and average acceptance length $\tau$ on AIME24.

| Models | Method | Speedup | $\tau$ |
|---|---|---|---|
| **DeepSeek-R1-Distill-Qwen-7B** | FAFO-Stream | **1.60**× | 2.40 |
| | FAFO-Quest | 1.46× | 2.35 |
| | Lookahead | 1.37× | 2.26 |
| **DeepSeek-R1-Distill-Llama-8B** | FAFO-Stream | **1.65**× | 2.41 |
| | FAFO-Quest | 1.50× | 2.27 |
| | Lookahead | 1.48× | 2.40 |

---

[5]`https://huggingface.co/datasets/microsoft/SCBench`

## G.3 Robustness under Multi-turn Evaluation (Multi-IF)

Table 6: Speedup ratio and average acceptance length $\tau$ on Multi-IF.

| Models | Method | Speedup | $\tau$ |
|---|---|---|---|
| | FAFO-Stream | **2.02**$\times$ | 2.84 |
| Llama-2-7b-chat | FAFO-Quest | 1.76$\times$ | 2.76 |
| | Lookahead | 1.69$\times$ | 2.73 |

## G.4 Robustness under SCBench

Table 7: Speedup ratio and average acceptance length $\tau$ on datasets from SCBench.

| Models | Method | Math.Find | | RepoQA | | ICL.ManyShot | | Retr.MultiHop | |
|---|---|---|---|---|---|---|---|---|---|
| | | Speedup | $\tau$ | Speedup | $\tau$ | Speedup | $\tau$ | Speedup | $\tau$ |
| **L3.1 8B** | FAFO-Stream | 1.37$\times$ | 2.88 | 1.26$\times$ | 2.60 | 1.26$\times$ | 2.61 | 1.46$\times$ | 2.90 |

## G.5 More KV Cache Compression Method

We feature a third KV-cache compression method—SnapKV—in addition to Stream and Quest, which we have extensively benchmarked. This complementary design broadens the applicability of our approach across tasks and models.

Table 8: Speedup ratio and average acceptance length $\tau$ on **Llama3-8B-Instruct** with **FAFO-SnapKV**. Entries show *speedup*$\times$ ($\tau$).

| Models | Method | MTBench | HumanEval | Multi-IF |
|---|---|---|---|---|
| **Llama3-8B-Instruct** | FAFO-SnapKV | 1.69$\times$ (1.90) | 1.87$\times$ (2.41) | 1.81$\times$ (2.47) |

## G.6 FAFO vs. TriForce

Because TriForce's native implementation is incompatible with GQA, also to ensure that we are not running it in a much disadvantaged setting (as the 0.21$\times$ seems abnormal from a quick scan), we tested TriForce in its own reported setting (Yarn-Llama-2-7b-128k with PG-19 and NarrativeQA) in Table 10 and Table 9. In this setting, we do observe significant practical speedup from Tri-Force (though still below FAFOs) and 5$\times$+ of $\tau$, this hints two conclusions: 1) TriForce is most performant under easy general language modeling tasks, but not challenging, goal-specific tasks; and 2) Even under such tasks, TriForce's implementation bottlenecks from fully leverage its high guess generation quality, likely because of complexity of three-model pipeline. In contrast, FAFO is capable of offering decent improvement over all reported tasks.

Table 9: Speedup ratio and average acceptance length $\tau$ on **TriForce's NarrativeQA** with **Yarn-Llama-2-7b-128k**. Entries show *speedup*$\times$ ($\tau$).

| Model | Method / Input Length | 3072 | 5120 | 10240 |
|---|---|---|---|---|
| **Yarn-Llama-2-7b-128k** | TriForce | 1.44$\times$ (4.26) | 1.37$\times$ (4.12) | 0.29$\times$ (0.11) |
| | FAFO-Stream | **2.50**$\times$ (4.23) | **1.80**$\times$ (4.01) | **1.24**$\times$ (4.07) |

## G.7 Ablation Study

We do ablation study on different number of guesses and on the length of each $k$-gram guess. On MT-Bench (11), FAFO-Stream exhibits a clear interior optimum: short guesses ($k$=4) underperform

Table 10: Speedup ratio and average acceptance length $\tau$ on different context lengths of PG-19.

| Models | Method | 1024 | | 2048 | | 3072 | |
|---|---|---|---|---|---|---|---|
| | | Speedup | $\tau$ | Speedup | $\tau$ | Speedup | $\tau$ |
| Yarn-Llama-2-7b-128k | FAFO-Stream | **2.71×** | 3.03 | **1.90×** | 2.84 | **1.80×** | 2.65 |
| | FAFO-Quest | 2.03× | 2.38 | 1.35× | 2.10 | 1.22× | 1.93 |
| | Lookahead | 1.75× | 2.53 | 1.36× | 2.51 | 1.16× | 2.55 |
| | TriForce | 1.83× | 5.75 | 1.63× | 5.14 | 1.74× | 5.50 |

$(0.90\times, \tau{=}1.90)$, while mid-range guesses around $k{\approx}6\text{–}7$ yield the highest speedups ($\sim 1.9\times$) with only modest increases in acceptance length ($\tau{\approx}2.10\text{–}2.14$). Pushing $k$ beyond this range slightly tapers speedup (e.g., $k{=}8\text{–}9$: $1.75\text{–}1.81\times$) without $\tau$ benefits, indicating diminishing returns once guesses get too long.

Table 11: Speedup ratio and average acceptance length $\tau$ on different lengths of $k - gram$ guess subsequences on MT-Bench.

| Models | Method | 4 | | 5 | | 6 | | 7 | | 8 | | 9 | |
|---|---|---|---|---|---|---|---|---|---|---|---|---|---|
| | | Speedup | $\tau$ | Speedup | $\tau$ | Speedup | $\tau$ | Speedup | $\tau$ | Speedup | $\tau$ | Speedup | $\tau$ |
| **Llama-2-7b-chat** | FAFO-Stream | 0.90× | 1.90 | 1.03× | 2.04 | 1.90× | 2.10 | 1.90× | 2.14 | 1.75× | 2.20 | 1.81× | 2.15 |

Varying the number of parallel guess subsequences (12) consistently increases speedup: FAFO-Stream improves from $\sim 1.05\times$ at 10 to $\sim 1.92\times$ at 40, and $\tau$ rises smoothly from $\sim 1.9$ to $\sim 2.3$. FAFO-Quest follows the same monotonic trend.

Table 12: Speedup ratio and average acceptance length $\tau$ on different number of guess subsequences.

| Models | Method | 10 | | 20 | | 30 | | 40 | |
|---|---|---|---|---|---|---|---|---|---|
| | | Speedup | $\tau$ | Speedup | $\tau$ | Speedup | $\tau$ | Speedup | $\tau$ |
| **Llama-2-7b-chat** | FAFO-Stream | 1.05× | 1.92 | 1.78× | 2.10 | 1.81× | 2.20 | 1.92× | 2.30 |
| | FAFO-Quest | 0.90× | 1.86× | 1.59× | 2.10 | 1.64× | 2.17 | 1.70× | 2.19 |

Finally, we conduct compression ratio ablation study (Table 13) on Multi-IF with Llama-2-7b-chat. Result shows a familiar trade-off: a moderate Init+Local token budget ($\approx 760$ tokens) maximizes speedup ($2.02\times$) while keeping $\tau$ flat ($\approx 2.84$), whereas overly aggressive compression (1360) lowers speedup ($1.59\times$).

Table 13: Speedup ratio and average acceptance length $\tau$ on **Multi-IF** with **Llama-2-7b-chat** across different compression settings. Entries show $speedup\times$ ($\tau$).

| Model | Method / Init+Local Tokens | 360 | 560 | 760 | 1360 |
|---|---|---|---|---|---|
| **Llama-2-7b-chat** | FAFO-Stream | 1.93× (2.76) | 1.98× (2.83) | **2.02×** (2.84) | 1.59× (2.86) |

# H   KV CACHE MANAGEMENT

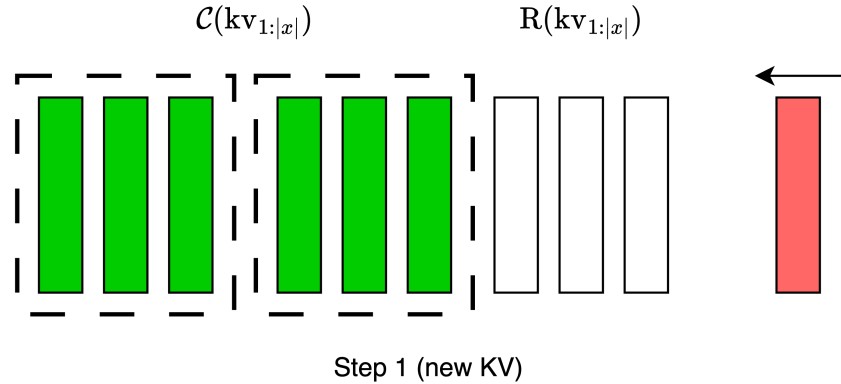

Step 1 (new KV)

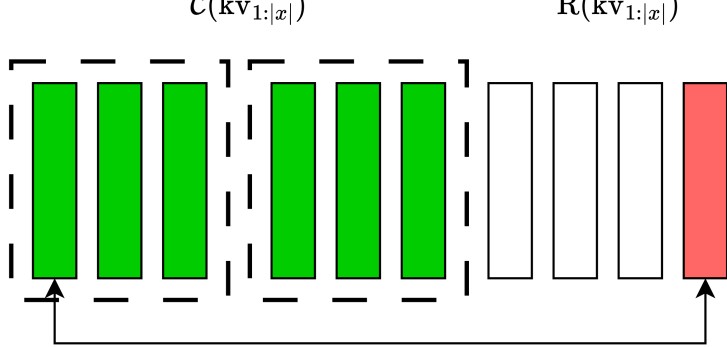

Step 2 (swap selected KV)

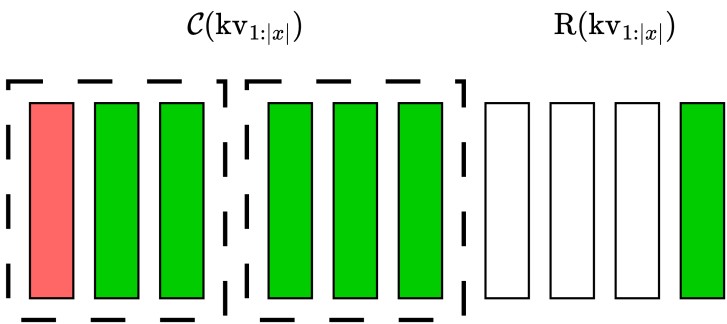

Step 3 (update new KV cache block for Fumble Decoding)

Figure 5: KV Cache Management with selective swapping into fixed-size KV blocks. The red KV cache represents the newly generated entry at the current decoding step, while the green KV caches denote previously selected entries retained by the compression function $\mathcal{C}$.

In Figure 5, suppose we are using StreamingLLM as the KV cache compression function. After each decoding step, new KV cache entries are generated (Step 1). FAFO manages the cache by swapping the new KV entries with discarded ones—the oldest ones in this case, utilizing StreamingLLM as function $\mathcal{C}$. This operation takes place within a fixed region of KV cache blocks that is shared by both the Fumble Decoding and Find Out phases, ensuring that all relevant KV entries remain compactly located.

This design is well-suited to FlexAttention, which improves efficiency by skipping over blocks that are entirely sparse Dong et al. (2025). Since FAFO maintains the active KV entries within a fixed number of blocks (two blocks in Figure 5), FlexAttention only needs to load these specific blocks into the GPU's streaming multiprocessors (SMs) for attention computation. In contrast, without FAFO, the KV cache becomes scattered across all cache blocks, forcing FlexAttention to load the entire KV cache, even when most blocks contain only discarded or irrelevant entries. A visual comparison is shown in Figure 3.

Moreover, we would like to emphasize that the *cost of precomputing the BlockMask for FlexAttention is prohibitively high*, rendering it impractical in the context of the dynamic and ever-growing KV cache during the autoregressive inference process. While one might argue that in methods such as STREAMLLM, where the number of KV cache blocks remains constant, FLEXATTENTION could similarly load only a small number of blocks, this overlooks a critical limitation: maintaining this efficiency would require *recomputing the BlockMask at every decoding step* to reflect the current structure of the compressed KV cache. Unfortunately, this recomputation is **extremely expensive— often more costly than simply loading the full KV cache into memory**. As a result, without a mechanism like FAFO to ensure locality and block consistency over time, the theoretical benefits of sparse attention are outweighed by the overhead of managing the sparsity structure itself.

---

**Algorithm 1** FAFO-Optimized FlexAttention Kernel (Block-Parallel)

---

**Require:** Query $Q$, Key $K$, Value $V$ in HBM
**Require:** Fixed Block Budget $N_{active}$, Block Size $B_z$
**Require:** Grid Dimension $M = \lceil \text{seq\_len}/B_z \rceil$
 1: **// Kernel Grid Launch: Each Thread Block handles one Query Tile**
 2: **for all** Query Block index $i \in \{0, \dots, M-1\}$ **in parallel do**
 3:     Load $Q_i = Q[i \cdot B_z : (i+1) \cdot B_z]$ from HBM $\rightarrow$ SRAM
 4:     Initialize accumulator $O_i \leftarrow 0$
 5:     **for** $j = 0$ to $N_{active} - 1$ **do**
 6:         **// Load Key/Value Blocks**
 7:         Load $K_j, V_j$ from HBM address $[j \cdot B_z] \rightarrow$ SRAM
 8:         **// Block-wise Attention Computation**
 9:         $S_{ij} \leftarrow Q_i \cdot K_j^T$
10:         Apply logical score mod: $S_{ij} \leftarrow S_{ij} + M_{logic}[i, j]$
11:         **// Update Output Accumulator**
12:         $O_i \leftarrow \text{Update}(O_i, S_{ij}, V_j)$
13:     **end for**
14:     **// 3. Writeback Result**
15:     Write $O_i$ from SRAM $\rightarrow$ HBM address $[i \cdot B_z]$
16: **end for**
17: **Result:** Memory traffic reduced to $O(M \cdot N_{active} \cdot B_z)$

---

# I  VERIFICATION ON LOSSLESS GENERATION QUALITY OF FAFO

Theoretically proven in Leviathan et al. (2023), FAFO preserves lossless generation quality. Following Lookahead Decoding's evaluation practice, we benchmark on LLaMA-2-7B-Chat over 160 samples from MT-Bench using Hugging Face greedy as the reference. Under FP3*, FAFO reproduces the greedy outputs exactly on 157/160 samples, whereas only has difference from 3 - 10 characters in the remaining cases. Under FP16, FAFO's outputs align with the corresponding Hugging Face greedy results, while exhibiting small deviations from the FP32 reference (36/160). Thus, although practical runs at reduced precision may not perfectly match Hugging Face greedy, FAFO retains the greedy output distribution within the numerical error range—no worse than Hugging Face's half-precision behavior—while remaining lossless by construction under identical numerical settings.

## J "FIND OUT" CACHING, RETRIEVAL, AND VERIFICATION ALGORITHM

---

**Algorithm 2** "Find Out" Caching

---

**Input:** A cache pool $G$, $n$ subsequences $y^1_{s_1+2:s_1+k+1}, \cdots, y^n_{s_n+2:s_n+k+1}$, and their buffers of discarded tokens $y^1_{s_1-k:s_1+1}, \cdots, y^n_{s_n-k:s_n+1}$
**Output:** Updated cache pool $G$
**for** $i = 1$ **to** $n$ **do**
    **for** $j = s_1 + 1$ **down to** $s_1 - k$ **do**
        $G[(y^1_{j:s_1+1})].\text{add}(y^i_{s_i+2:s_i+k+1})$
    **end for**
**end for**
**return** $G$

---

**Algorithm 3** "Find Out" Retrieval

---

**Input:** An input sequence $x_{1:|x|}$, a cache pool $G$, number of sequence to be retrieved for verification $m$, sequence length $k$
**Output:** $m$ retrieved sequences for verification
{{} denotes a set}
retrievedSeqs $\leftarrow$ {}
suffixSeqLen $\leftarrow k + 2$

**while** size(retrievedSeqs) $< m$ & suffixSeqLen $> 0$ **do**
    **for** sequence $y \in G[x_{|x|-\text{suffixSeqLen}:|x|}]$ **do**
        retrievedSeqs $\leftarrow$ retrievedSeqs $\cup$ $y$
        **if** size(retrievedSeqs)$= m$ **then**
            **Break**
        **end if**
    **end for**
**end while**
**return** retrievedSeqs

---

**Algorithm 4** "Find Out" Verification

---

1: **Input:** An input sequence $x_{1:|x|}$, a language model $p$, $m$ candidate sequences $A = \{a^1_{1:k}, \ldots, a^m_{1:k}\}$
2: **Output:** accepted tokens $c$
3: $c \leftarrow \emptyset$
4: **for** $a \in A$ **do**
5:     $\mathcal{D} \leftarrow p(x_{|x|}|x_{1:|x|-1})$
6:     $c' \leftarrow \emptyset$
7:     **for** $i = 1$ **to** $n$ **do**
8:         **if** argmax($\mathcal{D}$) = $a_i$ **then**
9:             $\mathcal{D} \leftarrow p(a_i|c', x_{1:|x|})$
10:             $c' \leftarrow [c'|a_i]$
11:         **else**
12:             **break**
13:         **end if**
14:     **end for**
15:     $c' \leftarrow [c'|\text{argmax}(\mathcal{D})]$
16:     **if** size($c'$) $>$ size($c$) **then**
17:         $c \leftarrow c'$
18:     **end if**
19: **end for**
20: **return** $c$

---

