# OpenReview forum: "FAFO: Lossy KV Cache Compression for Lossless Inference Acceleration via Draftless Fumble Decoding"
_ICLR.cc/2026/Conference — Submitted to ICLR 2026_

### Official Review · Reviewer_xKQY · 2025-10-30

**Soundness:** 2
**Presentation:** 1
**Contribution:** 2
**Rating:** 2
**Confidence:** 4

**Summary:**

This paper proposes FAFO (Fumble Around and Find Out), a lossless LLM inference acceleration framework based on combining KV-cache compression with lookahead decoding. The KV-cache compression method used in FAFO is StreamingLLM or Quest, while draftless fumble decoding is presented to realize lookahead decoding. Empirical results show a latency speedup of 1.20-2.71x on a set of Llama and Qwen LLMs.

**Strengths:**

1. The idea of combining KV-cache compression with lookahead decoding (draftless fumble decoding) looks novel and interesting.
2. A latency speedup of 1.20-2.71x over the original model is significant.

**Weaknesses:**

1. This paper is poorly structured and written, very verbose, and difficult to follow. The writing quality hinders fair evaluation of the technical content. I suggest the authors completely rewrite and resubmit the paper -- the technical content and contribution could not be fairly evaluated in the current shape.
2. A latency speedup of 1.20-2.71x over the original model is significant. However, if the "Lookahead" case is considered as the baseline, the speedup will be much less. In addition, it is unclear how much of the speedup comes from FlexAttention -- this should not be regarded as FAFO's contribution.

**Questions:**

Please see "Weaknesses" for my major comments and suggestions.
1. From Weakness 1: The very first issue of this paper is the writing. Please fix it.
2. From Weakness 2: Should the baseline be the original model, or the "Lookahead" case?
3. From Weakness 2: How much of the speedup comes from FlexAttention?
4. Lack of ablation studies on lookback window size
5. Missing related work: LongSpec

---

> ### Author Response · Authors · 2025-11-24
> **Thanks, and we lookforward to your feedback (1/2)**
>
> ### **`W1, Q1 - presentation.` We hear you, but presentation is often a multifaceted subject where "one man's vulgarity is another's lyric." Here, we present some updates and please see if they are helpful.**
>
>
> The reviewer didn't give much detail on what exactly made our paper *“poorly structured and written,”* other than calling it *“very verbose.”* We consider ourselves fairly experienced paper writers — our submissions are often praised by reviewers. In fact, for this same paper, reviewer `Cruo` lists the following strength:
>
> > `Cruo` Clear presentation: The paper clearly contrasts FAFO with prior speculative decoding methods (TriForce, Lookahead, Self-SD). Figure 2 effectively illustrates the mechanism, making the concept easy to grasp.
>
> This seems to contradict the reviewer’s feedback. And we can further disclosed that in our last submission, reviewer find a strength of work to be:
>
> > The paper is clearly written and easy to follow.
>
> ---
>
> That said, other reviewers also found our presentation improvable. For instance, reviewers `qBqd` and `amQi` both found Section 1.1 (failure modes of lossy KV cache compression) to be overly long, likely resonating with the “very verbose” concern of yours.
>
> **We agree with this criticism and fixed such writing. But at the same time, please allow us to present some backstory:** In fact, much of our drive to write this longer-than-necessary Section 1.1 is due to our observation of DFloat11, which is another piece of recent lossless compression work that has received great community interest. By inspecting the [online discussions](https://www.reddit.com/r/MachineLearning/comments/1k7of6w/rp_we_compress_any_bf16_model_to_70_size_during/) of DFloat11 and its [reviews at NeurIPS](https://openreview.net/forum?id=xdNAVP7TGy), **it is clear that many members do not see the value of lossless compression unless failure modes of lossy methods are explicitly spelled out.** We can even disclose that the last time we submitted this paper, we received criticism that our verification step still requires full cache (which is a clear sign of not seeing the value of lossless quality) and that n-gram decoding is just like SD and therefore our contributions are trivial  (which demonstrates a clear lack of efficient decoding background), thus the elaborated Section 1.1 and other background sections.
>
> Similarily, while `qBqd` prefers more system-level details, this preference is in clear contrast to reviewer `amQi`'s feedback, who believes that if more system details are added, we should not submit to ICLR but to a more systems-focused venue. While both `qBqd` and `amQi` want less background, the above-quoted reviewer `Cruo` finds "clear presentation" to be a strength of our work and appreciates how we contrast FAFO to prior works.
>
> ---
>
> We list these examples and anecdotes not to discount your feedback, but to illustrate the point that **presentation is often a multifaceted subject with no universally good answer; *"one man's vulgarity is another's lyric"* is very much the name of the game given the growing size of ML community.** On this note, we ask for reviewer leniency and patience when encountering information you are familiar with, as this is likely due to your experience and expertise, while the general community (or another expert of different focuses) might find such basics useful.
>
> That said, **we again agree that many improvements can be made in terms of organization. We have now:**
>
> * Reduced Section 1.1 and moved failure mode details to the Appendix.
> * Added paragraph highlights in Section 1.2 to better distinguish background, criticism (of self-SD), and motivation (for why n-gram).
> * Modified end-of-introduction contribution bullet points to better distinguish “contribution over self-SD” and “contribution over Lookahead.”
> * Reduced repetition of the n-gram paradigm in Section 3.
> * Added a clearer roadmap in Section 3 leading up to the investigations in Sections 3.1 and 3.2.
> * Significantly expanded system-level details in Section 4.
> * Added a limitation section as Appendix B to highlight that FAFO currently lacks support for batched inference.
>
> ---
> We hope these updates help and would allow the reviewer to give our work a proper evaluation. **We venture to argue that it would be too extreme to claim that**
>
> > `xKQY` the technical content and contribution could not be fairly evaluated in the current shape
>
> **as our work has already been evaluated on its technical merits for seven independent reviewers** (three others this time, and four in our previous submission). We look forward to your feedback.

---

> ### Author Response · Authors · 2025-11-24
> **Thanks, and we lookforward to your feedback (2/2)**
>
> ### **`W2.1, Q2 - Speedup over original model or Lookahead.` The reported speedups are over the original model under vanilla decoding, as is standard practice in the field and specified in many parts of our work.**
>
> We have explicitly declared and defined the speedup readings are over the original model in many parts of our work, for instance:
>
> > `Abstract, L031` Our comprehensive empirical results show FAFO provides 1.20-2.71$\times$ latency speedup over the original model
>
> > `Section 5, L486` **Wall-clock speedup ratio**: The observed speedup relative to vanilla autoregressive decoding, measured in tokens per second.
>
> To the best of our knowledge, this is the absolute standard practice in similar literature. For instance, the reviewer-referenced LongSpec claims:
>
> > 2.25× reduction in wall-clock time on the AIME24 long reasoning task with the QwQ model
>
> which is also over the original model, even though it features other SD methods (e.g., MagicDec) as baselines.
>
> As for the core question — how FAFO compares to Lookahead from a performance perspective — the answer is: quite a lot. Lookahead is featured extensively in our experiments, and it can be observed that FAFO consistently beats Lookahead in both speedup and average acceptance length ($\tau$). The gap can be as approximately 30% and as significant as +1.35× (Table 3).
>
> ---
>
> ### **`W2.2, Q3 - How much of the speedup comes from FlexAttention?` None, because FlexAttention is slower than FlashAttention.**
>
> There are two potential ways to interpret this question:
>
> 1. How much speedup does FlexAttention provide over the standard attention implementation (FlashAttention)?
> 2. How much speedup gap is there between FAFO with Flex vs. FAFO without Flex?
>
> Under the first interpretation: **FlashAttention** — which is the standard kernel used in vanilla model decoding — **is actually faster than FlexAttention**, with Flex designed primarily for enhanced customizability. This is evidenced by the following [quote](https://pytorch.org/blog/flexattention/) and results from Flex’s authors:
>
> > `Performance` *"FlexAttention achieves 90% of FlashAttention2’s performance in the forward pass and 85% in the backward pass"*
>
> In other words, FAFO intentionally accepts the cost of a slower attention kernel in exchange for the masking customizability required to implement the custom KV cache management system essential to FAFO.
>
> Under the second interpretation: this is impossible to test, because FAFO *requires* Flex to maintain the masks necessary for its guess generating design to function. There is no configuration of "FAFO without Flex" unless one develops custom kernels specifically for it (and it would be for one particular type of FAFO — e.g., FAFO-Stream). That said, there are two pieces of information that may address the reviewer’s broader intent:
>
> 1. Without our custom KV cache management scheme, a brute-force integration of FAFO’s ideas into FlexAttention would not work, as it would require mask reconstruction at every step — causing massive slowdowns.
> 2. If one were to hardcode FAFO’s attention implementation directly into a custom kernel, this would indeed produce a “FAFO-Stream without Flex.” And with careful kernel engineering, such an implementation would generally outperform "FAFO-Stream with Flex" from an end-to-end perspective — since custom kernels (as in this hypothetical FAFO-Stream) are typically faster than general-purpose kernels (such as Flex).
>
> We hope such info are helpful.
>
> ---
>
> ### **`Q4 - Ablation study on lookback window size` Sure, here it is.**
>
> We conduct an ablation study using 9-gram, 20 guesses generated, and 12 guesses retrieved from the cache pool for verification per forward step. The results are shown below. As the lookback window increases, the model retrieves guesses that are more likely to be verified correctly, which in turn yields additional speedup.
>
> > Different lookback window with Llama-2-7b-chat on MT-Bench
>
> | lookback window length | Speedup | $\tau$ |
> |-|:-:|:-:|
> | 3  | 1.89× | 2.28 |
> | 5  | 1.94× | 2.29 |
> | 7  | 1.97× | 2.29 |
> | 10 | 1.92× | 2.28 |
>
> ---
> ### **`Q5 - Missing related work: LongSpec.`  LongSpec is not a draftless speculative decoding work, so it is incomparable.**
>
> We have, however, now included it in our Extended Related Works section in Appendix B, along with several other SD works focused on long context settings.

---

### Official Review · Reviewer_amQi · 2025-10-31

**Soundness:** 3
**Presentation:** 2
**Contribution:** 2
**Rating:** 4
**Confidence:** 3

**Summary:**

Authors present FAFO, an n-gram candidate-based decoding method for efficient token generation while also preserving full model quality. The Fumble Around step generates multiple n-gram candidate guesses using a compressed KV cache while the Find Out step verifies the candidates conditioned on a set of tokens. Both steps run in parallel, achieving draftless decoding. The work also includes customized cache managers built on FlexAttention kernels.

**Strengths:**

- The problem and proposed approach are very timely and needed in the current scenario. The ideas and implementations allow us to realize the benefits of n-gram candidate-based decoding methods in a practical setting.
- Authors present a complete engineering solution, implementing a custom KV cache manager with a smart memory layout (well explained in appendix) designed to work with a sparse attention kernel (FlexAttention). The system-level contributions, especially the fixed-size KV block design and swapping mechanism (Appendix G), address real implementation challenges that have prevented prior work from achieving practical speedups.
- Comprehensive empirical evaluation across multiple models (Llama-2, Llama-3, Llama-3.1, Qwen2.5), tasks, and settings. The experiments demonstrate consistent improvements and robustness across scenarios where baselines struggle (e.g., MT-Bench, long-context tasks).

**Weaknesses:**

**The Motivation and Introduction of the work feels all over the place**

- The title says, "Lossy KV Cache compression" then "Lossless Inference". Line 46, “Lossless KV cache compression framework". The abstract emphasizes “..lossy compression techniques can fumble..”.
  - Throughout the paper, the term *loss* conflates two concepts: (1) using lossy compression methods as a component within the system, and (2) the end-to-end generation quality being lossless. The introduction should clearly establish how FAFO uses lossy KV cache compression for candidate generation and maintains lossless output quality through verification similar to speculative decoding.
- Section 1.1 needlessly spends significant space (lines 72-101) articulating the lossy nature of lossy compression methods. While motivation is important, this is tangential to FAFO since it does not solve the compression problem. The key idea, “lossy methods can generate useful candidates even if they're not reliable for end-to-end generation” can be explained much more concisely.
- Section 1.2 first motivates how KV cache compression helps the already established SD paradigm by giving rise to SSD methods followed by their drawbacks. Authors then abruptly transition to “lossless efficient decoding channels” with n-gram candidate pool decoding established by Lookahead decoding. This is a jarring transition that leaves several questions unanswered:
  - Do SSD methods exist that use token-dropping KV cache compression? If yes, please discuss how (if any) they solve general drawbacks of SSD.
  - Does there already exist an efficient implementation that realizes the gains from above? Does Table 1 use it?
- Overall the limitations of SSD+compression approaches should be established **before** introducing n-gram methods as an alternative.
- Furthermore, the distinction between n-gram methods and SD as different paradigms (Line 199: "..given the parallel draft-and-verify vs the sequential draft-then-verify difference..") is crucial but introduced too late. Table 1 underscores the effectiveness of the n-gram candidates over SSD, but by this point we are already beyond introduction. Kindly make it clear at the start.
- From my understanding, A possible reframing:
  - Lossy compression enables efficient candidate generation but cannot be used end-to-end
  - Self-SD addresses this end-to-end by using compression for drafting + full cache for verification but faces memory and efficiency issues.
  - N-gram candidate pool decoding offers an alternative paradigm that can overcome these limitations through parallel verification.
  - However, integrating compression with n-gram methods is non-trivial and hasn’t been done before.



**Unclear Positioning relative to existing solutions and possibly overstated claims**

- Now, as the n-gram based methods are in a different paradigm than SSD, all the benefits provided by FAFO should **primarily** be compared with Lookahead decoding rather than with SSD methods. The fundamental properties, lossless output, single KV cache, parallel verification, etc follow directly from the “draft-and-verify vs draft-then-verify” paradigm rather than FAFO’s specific contributions. **Authors should clarify this** and avoid overstating claims.
  - Line 157, “..FAFO is the only approach capable of delivering such a trifecta[lossless quality, latency improvements, memory footprint] other than Lookahead Decoding..”  This is a bit misleading since these properties are inherent to the n-gram paradigm(and hence Lookahead decoding) and not unique contributions of FAFO.
  - The extensive comparison with TriForce is helpful for showing practical superiority but the paper should be clearer that these advantages come from the paradigm rather than algorithmic innovations.
- A clearer framing would be expanding the contribution point 4, where authors correctly identify that this work unlocks the capabilities from a stagnant paradigm by making n-gram based decoding practical and effective rather than saying “we solve SSD’s problems by using n-gram methods”


**Limited Conceptual Novelty beyond existing Paradigms**

- As noted in the previous points, conceptually the additional improvements to Lookahead decoding include:
  - Fumble Decoding (Using compressed KV-cache to generate candidate n-grams):  While this is extremely non-trivial from an implementation standpoint, conceptually it does not stand out. The key idea that compressed caches could generate useful candidates has been explored in previous SSD contexts.
  - Find out Verification (Retrieving candidates based on a longer suffix of tokens): This key innovation (Section 4.3) alone does not represent substantial conceptual advancement.

From a research perspective these ideas do not contribute substantially to existing work.
The real achievements (as noted in Strengths) are in engineering: Making FlexAttention work for dynamic decoding through fixed size block allocation, The swapping mechanism, Custom KV Cache manager. While these are certainly non-trivial accomplishments, the work would greatly benefit from either, (a) identifying additional algorithmic or theoretical insights beyond, “use compression in the n-gram paradigm” or (b) reframing as primarily systems contribution and submitting to an appropriate venue. As currently positioned, the conceptual contribution feels thin despite the solid engineering work.

**Questions:**

See Weaknesses

---

> ### Author Response · Authors · 2025-11-24
> **Thanks! (1/4)**
>
> Before rebutal, we want to thank the reviewer for presenting not only detailed feedback, but also taking the time to lay out an actionable plan for us to fix. That shows dedication that we don’t often find, and we applaud you for that.
>
> ---
>
> ### **`W1.1 - Lossy vs Lossless KV cache.` The reviewer is right that this term consistency is not ideal, fixed and thanks for the close read.**
>
> We have updated the phrase involving “lossless KV cache compression framework’’ to “In this work, we present **a framework that maintains lossless generation quality while using lossy KV cache compression as its means.**” This more clearly reflects that the compression itself is lossy while the resulting generation remains lossless. We thank the reviewer for the close read and for helping us fix a potentially confusing point from the very beginning of this paper.
>
> A similar mention of "lossless KV cache compression" appears at the end of Section 1.2 (contribution summary), which is also fixed for consistency.
>
> ---
>
> ### **`W1.2 - The key idea, “lossy methods can generate useful candidates even if they're not reliable for end-to-end generation” can be explained much more concisely.` We agree and now fixed!**
>
> We have now massively reduced Section 1.1. In our defense, much of our drive to write this longer-than-necessary Section 1.1 is due to our observation of DFloat11, which is another piece of recent lossless compression work that has received great community interest. By inspecting the [online discussions](https://www.reddit.com/r/MachineLearning/comments/1k7of6w/rp_we_compress_any_bf16_model_to_70_size_during/) of DFloat11 and its [reviews at NeurIPS](https://openreview.net/forum?id=xdNAVP7TGy), it is clear that many members do not see the value of lossless compression unless failure modes of lossy methods are explicitly spelled out. We can even disclose that the last time we submitted this paper, we received criticism that our verification step still requires full cache (which is a clear sign of not seeing the value of lossless quality), thus the elaborated Section 1.1.
>
> **That said, we agree that such failure mode examples probably shouldn't take that much space in the main text, and we have now reduced Section 1.1 and moved more examples to Appendix C.** We thank you and reviewer `qBqd` to highlight this issues.
>
>
> ---
> ---
>
> ### **`W1.3 - SSD should be established before introducing n-gram as alternative.` Respectfully, this is exactly what we did and there might be a misread.**
>
> ## **`We believe there is likely a significant misread of Section 1.2 of our work. In fact, we find our presentation to be exactly in line with the reviewer-recommended reframing.` Please allow us to walk through it in detail.**
>
> (For easier reading, we use `old line number | current line number` to mark quotes, where the former indicates the line number of our PDF at submission — which, to our understanding, is not available to reviewers once an revision is posted — and the latter indicates the line number of current PDF.)
>
> ---
>
> The reviewer first notes the transition between SSD and n-gram is "jarring," and asks:
>
> > * Do SSD methods exist that use token-dropping KV cache compression? If yes, please discuss how (if any) they solve general drawbacks of SSD.
>
> In fact, we have listed many SSD methods incorporating KV cache compression, which directly addresses the question of *"Do SSD methods exist that use token-dropping KV cache compression?"*
>
> > `L113 | L103` A subfield named Self-Speculative Decoding (self-SD) has since been promoted... Specifically, **methods like Sun et al. (2024); Sadhukhan et al. (2025); Zhang et al. (2023a); Xia et al. (2025) have explored the general idea of integrating lossy KV cache compression with speculative decoding**, attempting to deliver lossless generation quality with improved latency.
>
> And immediately after this paragraph, we walk through their shortcomings (some are directly attributed to such SSD methods), addressing the request of *"If yes, please discuss how (if any) they solve general drawbacks of SSD."*
>
> > `L119 | L113` However, **we find all such self-speculative decoding methods come with two practical shortcomings.** First, they all require maintaining separate sets of ... Secondly, we find many such self-SD methods lack general usability: for instance, TriForce Sun et al. (2024) demands... SS Zhang et al. (2023a) can require... MagicDec Sadhukhan et al. (2025) only performs well under...

---

> ### Author Response · Authors · 2025-11-24
> **Thanks! (2/4)**
>
> As for the reviewer’s second question:
>
> > * Does there already exist an efficient implementation that realizes the gains from above? Does Table 1 use it?
>
> Table 1 is a small pilot study to support the design of FAFO, but not a comprehensive evaluation table. That said, TriForce is indeed already featured in Table 1 (we now also include SWIFT there). And methods like TriForce and SWIFT are included as baselines in Table 2 (one of our main eval tables) and more. We also explicitly discussed why the two other works (MagicDec and SS) are omitted around `L431 | L494`.
>
> ---
>
> The reviewer further notes:
>
> > Overall the limitations of SSD+compression approaches should be established before introducing n-gram methods as an alternative.
>
> Which is exactly what we did. We first introduced many SSD + KV cache compression methods at  `L115 | L108`, then discussed their shortcomings at `L119 | L113`. The n-gram methods are introduced as an alternative **after** these two paragraphs, around `L132 | L128`.
>
> ---
>
>
> ### **Here, we give a walkthrough of how your suggested reframing of `W1.3` actually matches our existing presentation.**
>
> We present your reframing suggestions as the bullet points, and match our corresponding presentation as subpoints, with line numbers to indicate order.
>
>
> ***"From my understanding, A possible reframing:"***
>
> * ***"Lossy compression enables efficient candidate generation but cannot be used end-to-end"***
>     * This is what our Section 1.1 does at `L094 | L086`, with quotes like:
>     * > `L094 | L086` While KV cache compression techniques bring significant efficiency benefits, direct deployment of models utilizing lossy compressed KV cache is often done at the cost of decreased model reliability.
>     * At the start of Section 1.2, we also reemphasize:
>     * > `L106 | L096` While lossy KV cache compression methods might have their own pitfalls if employed in an end-to-end manner, it is common knowledge that...
>
> * ***"Self-SD addresses this end-to-end by using compression for drafting + full cache for verification but faces memory and efficiency issues."***
>     * In Section 1.2, we first establish that self-SD is lossless and draftless and therefore has end-to-end potential.
>     * > `L113 | L103` A subfield named Self-Speculative Decoding (self-SD) has since been promoted... Specifically, methods like ... have explored the general idea of integrating lossy KV cache compression with speculative decoding, attempting to deliver lossless generation quality...
>     * Then, we point out their shortcomings.
>     * > `L119 | L113` However, we find all such self-speculative decoding methods come with two practical shortcomings...
>
> * ***"N-gram candidate pool decoding offers an alternative paradigm that can overcome these limitations through parallel verification."***
>     * This is also how we introduced n-gram in Section 1.2.
>     * > `L132 | L128` With this in mind, we look into other lossless efficient decoding channels and find the n-gram candidate pool paradigm pioneered by Lookahead Decoding (Fu et al., 2024) to be a potential candidate...  completely sidestep the first shortcoming mentioned above...
>
> * ***"However, integrating compression with n-gram methods is non-trivial and hasn’t been done before."***
>     * We establish this around:
>     * > `L138 | L136` However, n-gram decoding has its own quirks. Most significantly, since drafting and verification occur within the same forward pass, it requires non-trivial system engineering efforts to support...
>
> **It can be seen that our initial writing already followed the reviewer-recommended reframe to its exact order.**
>
> ---
>
> We note that, with a close read of Section 1.2, we have indeed identified three minor issues:
>
> * `L113 | L103` When we first introduce these KV-cache-compression–incorporated SD methods, we did not explicitly specify they are self-SD methods (even though the paragraph after does clarify this). We now explicitly mark them as self-SD methods.
> * `L138 | L136` We did not state that n-gram + lossy KV has not been done before immediately when introducing n-gram in Section 1.2; this is now explicitly claimed.
> * We hypothesize that part of the misread stems from Section 1.2 being a heavy section with the introduction of SD, self-SD, self-SD’s shortcomings, n-gram, and our contributions. We have now added paragraph titles and rewrite some connecting sentences to better guide readers.
>
> We hope the reviewer finds our walkthrough clear and our fixes helpful.

---

> ### Author Response · Authors · 2025-11-24
> **Thanks! (3/4)**
>
> ### **`W1.4 - Distinction between n-gram and SD are introduced too late.` We actually introduce it immediately after the mention of n-gram in Section 1.2.**
>
> The reviewer notes:
>
> > Furthermore, the distinction between n-gram methods and SD as different paradigms (Line 199: "..given the parallel draft-and-verify vs the sequential draft-then-verify difference..") **is crucial but introduced too late.** Table 1 underscores the effectiveness of the n-gram candidates over SSD, but by this point we are already beyond introduction. **Kindly make it clear at the start.**
>
> We respectfully note that we actually introduce this key distinction immediately after our mention of n-gram in Section 1.2.
>
> > `L132 | L128` With this in mind, we look into other lossless efficient decoding channels and find the n-gram candidate pool paradigm pioneered by Lookahead Decoding (Fu et al., 2024) to be a potential candidate. **Different from the sequential draft-then-verify design of SD, n-gram decoding generates its newly drafted n-grams in parallel with its lossless verification** (in Lookahead, both are done with full KV cache).
>
> We now highlight this in bold text.
>
> ---
> ---
>
>
> ### **`W2 - Unclear Positioning relative to existing solutions and possibly overstated claims.` We agree with the issues you pointed out, and the updated version clarifies FAFO's contribution over Lookahead vs over Self-SD methods.**
>
> The reviewer is observant in noting that we conflated FAFO's contributions over Lookahead vs over self-SD methods in our contribution summary. To clarify — other than the "typical" lossless claim — FAFO’s main contributions are:
>
> 1. Operate under leveled-memory footprint and a single KV cache.
> 2. Provide an interface via FlexAttention to connect lossy KV cache compression methods and the n-gram decoding paradigm.
> 3. Deliver general usability across a wide range of task scenarios.
> 4. Revive the n-gram paradigm via the Flex interface, enabling FAFO to host far more n-gram guesses than Lookahead (and also achieve noticeably better performance).
>
> Among them:
> * **#1** is an inherent property of n-gram and is not unique to FAFO; these are advantages of FAFO/Lookahead over Self-SD methods.
> * **#2** is unique to FAFO, so it is a contribution over Lookahead. It is technically also over self-SD, though self-SD methods typically don't need this interface to integrate with lossy KV cache compression method.
> * **#3** is mostly an advantage over Self-SD, though Lookahead does sometimes exhibit volatile performance (e.g., Table 3). We therefore believe it is fair to characterize this as a unique contribution of FAFO. We additionally note that most draftless + lossless decoding works do not evaluate as comprehensively as we did.
> * **#4** is, as the reviewer recognized, unique FAFO contributions over Lookahead. **Frankly, Lookahead is a method with many good properties but not good-enough performance; FAFO changes that and revive the lossless n-gram paradigm for future explorations.**
>
>
> We have updated our contribution summary to reflect these points clearly, please refer to `L152` of the new PDF. We thanks the reviewer for pointing out this issue.

---

> ### Author Response · Authors · 2025-11-24
> **Thanks! (4/4)**
>
> ### **`W3 - These ideas do not contribute substantially to existing work. The real achievements (as noted in Strengths) are in engineering.` We agree with this characterization that FAFO is integration/enginerring > algorithmic, but we argue it remains highly contributive within its research landscape.**
>
> We agree that our work largely adopts lossy KV cache compression under the n-gram context and lacks major algorithmic novelty. However, algorithmic novelty is only one part of a contribution, where the context of a work's respective landscape matters — please hear us out though some hypothetical Q&As:
>
> * **`Are there must-have constraints limiting the freedom of algorithmic novelty?` Yes — training-free + draftless + lossless decoding is highly practical but also extremely restrictive.**
>     * Common “novelty avenues” — custom training, architecture changes, draft/target co-design, lossy trade-offs — are off the table.
>     * Prior art under these constraints also reuses mature components: TriForce (COLM 2024), MagicDec (ICLR 2025), and SWIFT (ICLR 2025) all rely on integrating established KV compression with speculative decoding.
>
> * **`Is the field saturated with random repackaging?` No — there has been no lossless + draftless successor to Lookahead since its late-2023 debut.**
>     * If many lossy KV + n-gram methods existed, FAFO might look like random swapping and be of little value. But there has been zero lossless follow-up to Lookahead in two years, despite many lossy KV + SD proposals being proposed in the same period (see highlights by the end of Section 1). We believe it is fair to claim that FAFO practically revives the n-gram paradigm, as the reviewer also agreed.
>     * This suggests randomness doesn’t work — real progress needs deep understanding, careful component choice, and meticulous execution. We venture to argue FAFO delivers exactly that.
>
> * **`Is the performance gain massive?` Yes — FAFO’s speedup is exceptional.**
>     * The efficiency community values simplicity with impact. FAFO fixes Lookahead’s key drawback (unable to host more n-gram guesses) and delivers unmatched end-to-end acceleration under hard constraints across a broad task suite.
>
> * **`Rerouting known ingredients to solve a harder problem is a valid contribution.` FAFO meaningfully revives the stagnated n-gram candidate-decoding research space.**
>     * As the risk of being massively redundant, we highlight that many impactful efficiency works are algorithmically light but integration-savy:
>         * QLoRA = standard quantization + LoRA → foundation of memory-restrictive PEFT.
>         * KIVI = per-channel quantization + sliding-window buffer → foundation of KV-cache quantization.
>         * MEND = gradient decomposition + editing → foundation of large-scale LLM editing.
>     * We also believe the community would benefit more from having a FlexAttention interface enabling many lossy KV methods, than from us hardcoding a single unique method purely for scoring more novelty points.
>
> ---
>
> We are firm believers in one ARR guideline:
>
> > *“**The goal is to solve the problem, not to solve it in a complex way.** Simpler solutions are preferable — they are less brittle and easier to deploy.”*
>
> And we hope the reviewer shares this philosophy.
>
> ---
>
> As of the suggestion that we should either *“identify additional algorithmic or theoretical insights beyond ‘use compression in the n-gram paradigm’”* or *“reframe the work as primarily a systems contribution and submit to an appropriate venue.”* We pretty much did both, as we have
>
> * now updated the paper with more system details to address the feedback of reviewer `qBqd`.
> * added theoretical discussion in Appendix F to prove the monotonicity of verification success under longer matching prefixes.
>
> We are not too sure if the reviewer would now find our work “too system for ICLR.” At the risk of being redundant, we’d like to highlight **that many works whose main contributions are integration- or engineering-heavy are published at ICLR/ICML/NeurIPS.** For example, the aforementioned QLoRA, KIVI, and MEND were respectively accepted to NeurIPS, ICML, and ICLR. We also have works like FlexGen, DeepSpeed-MoE, TorchTitan, SGLang, etc. — which are objectively much more system-heavy than FAFO — being accepted to the same three ML conferences.

---

### Official Review · Reviewer_Cruo · 2025-11-01

**Soundness:** 3
**Presentation:** 3
**Contribution:** 3
**Rating:** 6
**Confidence:** 4

**Summary:**

The paper proposes FAFO (Fumble Around, Find Out), a speculative decoding method that integrates lossy KV-cache compression and verification within a single forward pass.
Instead of relying on a separate draft model, FAFO constructs an n-gram (typically 2-gram) candidate pool that predicts likely next tokens from the compressed KV cache.
Verification with the full KV cache occurs in parallel, enabled by a masked-attention design that allows both “fumble” (draft) and “find-out” (verify) computations in one pass.

**Strengths:**

+ **Interesting idea:** FAFO’s capability to perform n-gram drafting using a compressed KV cache and verify the n-gram drafts within the same forward pass is novel and technically elegant.

+ **Clear presentation:** The paper clearly contrasts FAFO with prior speculative decoding methods (TriForce, Lookahead, Self-SD). Figure 2 effectively illustrates the mechanism, making the concept easy to grasp.

**Weaknesses:**

+ **Limited workload consideration:** Experiments are limited to batch = 1. It remains unclear whether FAFO’s advantages persist under larger-batch or multi-sequence settings, where system-level bottlenecks may differ.

+ **Scalability concern:** FAFO introduces extra computation per decoding step. This additional work could shift the decoding regime from memory-bound to compute-bound, potentially diminishing speedups on larger batches or lower-end GPUs.

**Questions:**

+ **Scalability under larger batches:**
Have you evaluated FAFO with larger batch sizes or multi-sequence decoding? Since FAFO adds per-step computation and coordination, it would be helpful to understand whether the speedup persists as batch size increases or on GPUs with limited compute resources.

+ **Computation vs. memory trade-off:**
Can you quantify how much additional FLOPs FAFO introduces per decoding step compared to a standard n-gram (lookahead) or speculative decoding baseline? This would clarify when the approach transitions from memory-bound to compute-bound.

---

> ### Author Response · Authors · 2025-11-24
> **Thanks!**
>
> We thank the reviewer for recognizing our technical novelty and clear presentation. Honestly, the latter really comes in handy, as several other reviewers criticize us for being too heavy on background. While we respect all feedback, we believe such diverse reviews clearly illustrate how presentation is often a multifaceted subject where “one man’s vulgarity is another’s lyric;” and we aim to deliver our work in a considerate way for all audiences. Below, we address your raised concerns.
>
> ---
>
> ### **`W1, Q1 - Batched Inference.` Making n-gram decoding batch-supporting would likely require significant effort worthy of a work of its own.**
>
>
> The reviewer is again observant that FAFO is developed for latency-sensitive scenarios (batchsize = 1). Upon benchmarking — which is also actually by your request — we unfortunately cannot provide meaningful batched support at this time, as some challenging limitations appear baked into the current n-gram implementation. N-gram methods typically incur higher per-step FLOPs than vanilla decoding or SD methods, since they must perform both full-model verification and n-gram candidate generation within the same forward pass — making them far more likely to hit the compute-bound under batched settings. Such challenges suggest that a batch-capable n-gram design likely deserves a paper of its own.
>
> **We have now faithfully highlighted this shortcoming with more details in the Limitations section in Appendix B.** In our defense, SD methods that support meaningful batching and those that do not essentially fall into different categories, often requiring distinct designs and targeting different deployment scenarios (e.g., TriForce vs. MagicDec, even though both originate from the same lab). This distinction becomes especially salient when large batch inference is involved. It is our understanding that draftless methods make the most sense for private, resource-constrained deployments — settings where batching demand is low but latency and task versatility are prioritized.
>
> ---
>
> ### **`W2, Q2 - additional FLOPs introduced by FAFO and scalability concerns.` Sure — here we provide an ablation study on number of guesses that FAFO generates per step with the associated FLOPs measured.**
>
> We conduct an additional ablation study with 9-grams, retrieving 12 guesses from the cache pool for verification and varying the number of newly generated guesses per forward step. This configuration results in an additional  9 * (12 + # of guesses) tokens being processed per step. The experiment is run on an A100 GPU, and the results are reported below. We observe that the speedup peaks at around 20 new guesses and then decreases, indicating that the transition point — where the workload becomes compute-bound — occurs at roughly 20–30 new guesses (about 288–378 extra tokens per step).
>
>
> > TFLOPs/step and Speedup of different number of generated guesses per step of Llama-2-7b-chat on MT-Bench
>
>
> | \# guesses | TFLOPs/step | Speedup |
> |-|:-:|:-:|
> | 10 | 2.43 | 1.69x |
> | 20 | 3.38 | 1.89x |
> | 30 | 4.44 | 1.75x |
>
> As of comparing FAFO’s additional FLOPs with Lookahead and SD, we here provide a comparison with Lookahead and vanilla full model decoding.
>
> > Llama-2-7b-chat on MT-Bench
>
> | Method | TFLOPs/step | $\tau$ - Speedup (implementation overhead) |
> |-|:-:|:-:|
> | Vanilla | 0.02 | 0× (no overhead) |
> | Lookahead | 3.63 | (2.01–1.64)=0.37× |
> | FAFO-Stream | 3.38 | (2.25–1.91)=0.34× |
>
> It can be seen that efficient decoding methods often incur much higher FLOPs, yet still win on end-to-end latency under batchsize = 1. **The reviewer is correct in hypothesizing that this increased FLOPs consumption may cause difficulties at large batch sizes, though we believe this would be less of an issue on low-end GPUs,** as consumer-grade RTX 40 series cards typically offer respectable compute throughput (in comparison to the A100s we tested here) — with memory capacity and bandwidth (e.g., lack of SXM) being the primary limiting factors.
>
> *Two additional footnotes*
>
> (1) We additionally note that while the reviewer requested per-step FLOPs measurements for SD, there is no straightforward or aligned way to compare SD and n-gram methods in this regard. SD is inherently sequential, causing its FLOPs to fluctuate depending on which phase the method is in (draft vs verification), the draft length, and other implementation details. Here, we instead present $\tau$–Speedup to highlight the implementation efficiency of FAFO; while this metric does not reflect the same compute-bound transition illustrated in our first table, it provides an additional perspective for understanding FAFO’s overhead.
>
> (2) We are also not exactly sure if the reviewer is seeking a measured efficiency report or a theoretical analysis. In case the reviewer is more interested in a theoretical analysis of FAFO’s arithmetic intensity, please let us know and we are happy to provide one.

---

> ### Author Response · Authors · 2025-11-24
> **Not a weakness, but would you be so kind to check if our presentation updates won't lose you as an audience?**
>
> We have updated our presentation per the feedback of other reviewers. However, as our presentation feedback is highly mixed — with you finding our presentation clear, yet many others do not; and reviewers like `qBqd` and `amQi` seem to want opposite things — we want to ensure that these updates do not lose you as an audience.
>
> **Would you be so kind as to give our revised version a quick scan to confirm that the updated presentation is still reader-friendly for you?** Specifically, we have:
>
> * Reduced Section 1.1 and moved failure mode details to the Appendix.
> * Added paragraph highlights in Section 1.2 to better distinguish background, criticism (of self-SD), and motivation (for why n-gram).
> * Modified end-of-introduction contribution bullet points to better distinguish “contribution over self-SD” and “contribution over Lookahead.”
> * Reduced repetition of the n-gram paradigm in Section 3.
> * Added a clearer roadmap in Section 3 leading up to the investigations in Sections 3.1 and 3.2.
> * Significantly expanded system-level details in Section 4.
> * Added a limitation section as Appendix B to highlight that FAFO currently lacks support for batched inference.
>
> Thanks in advance.

---

### Official Review · Reviewer_qBqd · 2025-11-01

**Soundness:** 3
**Presentation:** 1
**Contribution:** 3
**Rating:** 6
**Confidence:** 4

**Summary:**

This paper makes a solid technical contribution to the speculative decoding domain. By unifying KV-cache compression with n-gram parallel verification under a single-KV, draftless design, FAFO advances the state of the art in both efficiency and practicality, especially for memory-constrained deployments such as local or on-device LLM inference. The work is conceptually well-motivated—aiming to reduce memory usage, preserve losslessness, and improve latency—and is empirically validated with consistent 1.2–2.7× speedups across multiple models and tasks.

However, the presentation and structure significantly weaken the paper’s impact. The writing often lacks focus, devotes excessive space to background discussion, and scatters key insights (particularly in Sections 1–3). Substantial portions are spent re-explaining the failure modes of lossy KV-cache compression and speculative decoding, even though the paper’s main contribution is not about memory savings from compression. Meanwhile, the core innovations—the parallel fumble–verify mechanism, KV-block management, and FlexAttention integration—are buried deep in the text and described with unnecessary verbosity. Overall, the paper’s technical clarity and organization fall short of top-tier standards. In my view, the framework would benefit from a more focused exposition that devotes space to the design rationale and engineering challenges behind its proposed solutions.

**Strengths:**

* Practical improvement to speculative decoding: single KV cache, no auxiliary draft model.

* Strong empirical results with robust gains across tasks and models.

* Careful engineering insight on attention sparsity and block-mask reuse.

**Weaknesses:**

* **Poor organization and writing quality:** The first 5–6 pages mix motivation, background, and criticism without a clear focus. The manuscript contains substantial redundancy; large portions of Section 1 repeat well-known discussions on the failure modes of lossy KV-cache compression. Due to limited high-level clarity, readers must piece together the core contributions from scattered descriptions, figures, and appendices.

* **Limited novelty in algorithmic ideas:** The core algorithmic concept is not particularly new — it essentially combines token sampling or KV-cache compression techniques with n-gram (Lookahead) decoding. The contribution lies more in integration than in conceptual innovation.

* **Insufficient explanation of system-level contributions:** While much of the technical advancement appears to stem from the system integration — such as the custom KV-cache manager and FlexAttention-based implementation — these components are described only at a high level, without sufficient technical depth or analysis to fully support the claimed efficiency improvements.

**Questions:**

* By default, FAFO offloads speculative subsequences to CPU memory and later reloads them to GPU memory. How is this offloading process managed in practice? Does it introduce noticeable latency or transfer overhead? Additionally, can FAFO operate in a fully GPU-resident (non-offloading) mode, and if so, how would that affect its performance and memory footprint?

* The evaluation only reports results with a batch size of 1. How does FAFO perform under larger batch sizes or even with a multi-GPU system? In particular, since the TriForce baseline shows improved throughput with batched inference, how does FAFO scale in comparison?

---

> ### Author Response · Authors · 2025-11-24
> **Thanks! (1/2)**
>
> Before the rebuttal, we want to extend our appreciation to the reviewer for your constrained review. It is honestly rare to find a reviewer who 1) appreciates the system-integration effort behind simple methods; and 2) is willing to give a positive rating even when the presentation is not fully to your tastes. We are glad to have you, and below, we try to address your feedback faithfully.
>
> ---
>
> ### **`W1, W3 - Poor organization and writing quality` We hear you and agree with many points, but presentation is often a multifaceted subject where "one man's vulgarity is another's lyric."**
>
> We agree with your criticism that the presentation of the paper can be improved. In particular, we agree that **Section 1.1 should be much reduced.** Though, in our defense, **we don't believe our highlighted failure modes of lossy methods are actually well-known.**
>
> In fact, much of our drive to write this longer-than-necessary Section 1.1 is due to our observation of DFloat11, which is another piece of recent lossless compression work that has received great community interest. By inspecting the [online discussions](https://www.reddit.com/r/MachineLearning/comments/1k7of6w/rp_we_compress_any_bf16_model_to_70_size_during/) of DFloat11 and its [reviews at NeurIPS](https://openreview.net/forum?id=xdNAVP7TGy), it is clear that many members do not see the value of lossless compression unless failure modes of lossy methods are explicitly spelled out. We can even disclose that the last time we submitted this paper, we received criticism that our verification step still requires full cache (which is a clear sign of not seeing the value of lossless quality) and that n-gram decoding is just like SD and therefore our contributions are trivial  (which demonstrates a clear lack of efficient decoding background), thus the elaborated Section 1.1 and other background sections.
>
> We believe the reviewer might find such failure modes and background well-known because you are an experienced expert in the field, much like you prefer more system-level details in `W3`. This preference is in clear contrast to reviewer `amQi`'s feedback, who believes that if more system details are added, we should not submit to ICLR but to a more systems-focused venue. Similarly, reviewer `Cruo` finds "clear presentation" to be a strength of our work and appreciates how we contrast FAFO to prior works.
>
> ---
>
> We list these examples and anecdotes not to discount your feedback, but to illustrate the point that **presentation is often a multifaceted subject with no universally good answer; *"one man's vulgarity is another's lyric"* is very much the name of the game given the growing size of ML community.** On this note, we ask for reviewer leniency and patience when encountering information you are familiar with, as this is likely due to your experience and expertise, while the general community (or another expert of different focuses) might find such basics useful.
>
> That said, we again agree that many improvements can be made in terms of organization. We have now:
>
> * Reduced Section 1.1 and moved failure mode details to the Appendix.
> * Added paragraph highlights in Section 1.2 to better distinguish background, criticism (of SSD), and motivation (for why n-gram).
> * Modified end-of-introduction contribution bullet points to better distinguish “contribution over SSD” and “contribution over Lookahead.”
> * Reduced repetition of the n-gram paradigm in Section 3.
> * Added a clearer roadmap in Section 3 leading up to the investigations in Sections 3.1 and 3.2.
> * Significantly expanded system-level details in Section 4.
> * Added a limitation section as Appendix B to highlight that FAFO currently lacks support for batched inference.
>
> We hope these updates help.
>
> ---
>
> ### **`Q1 - Non-offloading mode` We interestingly found that in fully GPU mode, FAFO (and Lookahead) is slower due to substantial tensor creation and concatenation.**
>
> In a single forward pass, all *Fumble Around* and *Find Out* guesses are concatenated with the current decoding token (see Figure 3 in Appendix E). Thus, regardless of whether we keep the guesses on the GPU or not, we must allocate a new tensor that contains all of these tokens. If we keep all guesses on the GPU, it’s worse, as we additionally need to materialize a new tensor for each new guess after every forward pass, which further increases decoding latency. Results below confirm this behavior for both FAFO and Lookahead.
>
> > Full GPU vs. CPU-offloading with Llama-2-7b-chat on MT-Bench
>
> | Baselines | Speedup | $\tau$ |
> |-|:-:|:-:|
> | FAFO (GPU) | 1.36x | 2.36 |
> | FAFO (CPU-offloading) | 1.91x | 2.29 |
> | Lookahead (GPU) | 1.27x | 1.66 |
> | Lookahead (CPU-offloading) | 1.61x | 1.66 |
>
>
> > Full GPU vs. CPU-offloading with Llama-2-7b-chat on HumanEval
>
> | Baselines | Speedup | $\tau$ |
> |-|:-:|:-:|
> | FAFO (full GPU) | 1.34x | 2.41 |
> | FAFO (CPU-offloading) | 2.03x | 2.34 |
> | Lookahead (full GPU) | 1.23x | 1.82 |
> | Lookahead (CPU-offloading) | 1.72x | 1.77 |

---

> ### Author Response · Authors · 2025-11-24
> **Thanks! (2/2)**
>
> ### **`W2 - The core algorithmic concept is not particularly new. The contribution lies more in integration than in conceptual innovation.` We agree with this characterization that FAFO is integration > algorithmic, but we argue it remains highly contributive within its research landscape.**
>
> We agree that our work largely adopts lossy KV cache compression under the n-gram context and lacks major algorithmic novelty. However, algorithmic novelty is only one part of a contribution, where the context of a work's respective landscape matters — please hear us out though some hypothetical Q&As:
>
> * **`Are there must-have constraints limiting the freedom of algorithmic novelty?` Yes — training-free + draftless + lossless decoding is highly practical but also extremely restrictive.**
>     * Common “novelty avenues” — custom training, architecture changes, draft/target co-design, lossy trade-offs — are off the table.
>     * Prior art under these constraints also reuses mature components: TriForce (COLM 2024), MagicDec (ICLR 2025), and SWIFT (ICLR 2025) all rely on established KV compression with speculative decoding.
>
> * **`Is the field saturated with random repackaging?` No — there has been no lossless + draftless successor to Lookahead since its late-2023 debut.**
>     * If many lossy KV + n-gram methods existed, FAFO might look like random swapping and be of little value. But there has been zero lossless follow-up to Lookahead in two years, despite many lossy KV + SD proposals being proposed in the same period (see highlights by the end of Section 1).
>     * This suggests randomness doesn’t work — real progress needs deep understanding, careful component choice, and meticulous execution. We venture to argue FAFO delivers exactly that.
>
> * **`Is the performance gain massive?` Yes — FAFO’s speedup is exceptional.**
>     * The efficiency community values simplicity with impact. FAFO fixes Lookahead’s key drawback (unable to host more n-gram guesses) and delivers unmatched end-to-end acceleration under hard constraints across a broad task suite.
>
> * **`Rerouting known ingredients to solve a harder problem is a valid contribution.` FAFO meaningfully revives the stagnated n-gram candidate-decoding research space.**
>     * As the risk of being massively redundant, we highlight that many impactful efficiency works are algorithmically light but component-savvy:
>         * QLoRA = standard quantization + LoRA → foundation of memory-restrictive PEFT.
>         * KIVI = per-channel quantization + sliding-window buffer → foundation of KV-cache quantization.
>         * MEND = gradient decomposition + editing → foundation of large-scale LLM editing.
>     * We also believe the community would benefit more from having a FlexAttention interface enabling many lossy KV methods, than from us hardcoding a single unique method purely for scoring more novelty points.
>
> ---
>
> We are firm believers in one ARR guideline:
>
> > *“**The goal is to solve the problem, not to solve it in a complex way.** Simpler solutions are preferable — they are less brittle and easier to deploy.”*
>
> And we hope the reviewer shares this philosophy.
>
>
> ---
>
> ### **`Q2 - Batched Inference.` Making n-gram decoding batch-supporting would likely require significant effort worthy of a work of its own.**
>
>
> The reviewer is again observant that FAFO is developed for latency-sensitive scenarios (batchsize = 1). Upon benchmarking, we unfortunately cannot provide meaningful batched support at this time, as some challenging limitations appear baked into the current n-gram implementation. N-gram methods typically incur higher per-step FLOPs than vanilla decoding or SD methods, since they must perform both full-model verification and n-gram candidate generation within the same forward pass — making them far more likely to hit the compute-bound under batched settings. Such challenges suggest that a batch-capable n-gram design likely deserves a paper of its own.
>
> **We have now faithfully highlighted this shortcoming with more details in the Limitations section in Appendix B.** In our defense, SD methods that support meaningful batching and those that do not essentially fall into different categories, often requiring distinct designs and targeting different deployment scenarios (e.g., TriForce vs. MagicDec, even though both originate from the same lab). This distinction becomes especially salient when large batch inference is involved. It is our understanding that draftless methods make the most sense for private, resource-constrained deployments — settings where batching demand is low but latency and task versatility are prioritized.
>
> Regarding TriForce, our experiments reveal many of its failure modes, including cases where it slows down to as low as 0.17× of the vanilla full model. From an end-to-end perspective, a batched TriForce configuration (reported to perform well at batchsize = 6) would likely still offer lower throughput than FAFO under many realistic task scenarios.

---

### Author Response · Authors · 2025-12-04
**Summary of reviewers' feedback and our rebuttals (1/2)**

Dear AC triplets,

Despite some semi-decent initial scores (`6642`), our work has unfortunately not received any reviewer engagement. So much of the pressure would fall on the AC. To facilitate, here we provide a concise summary of the reviewers’ recognitions, followed by a summary of our rebuttal.


---
# **Reviewers' Recognitions**


We are glad to report that


- **Reviewers acknowledge the strong performance of our method, supported by rigorous benchmarking across diverse datasets:**
	- `qBqd`:  *"Strong empirical results with robust gains across tasks and models."*
	- `qBqd`:  *"FAFO advances the state of the art in both efficiency and practicality, especially for memory-constrained deployments such as local or on-device LLM inference."*
	- `amQi`:  *"Comprehensive empirical evaluation across multiple models, tasks, and settings. The experiments demonstrate consistent improvements and robustness across scenarios where baselines struggle..."*
	- `xKQY`:  *"A latency speedup of 1.20-2.71x over the original model is significant."*
- **Reviewers acknowledge the significant system-level contribution of our work:**
	- `qBqd`:  *"Careful engineering insight on attention sparsity and block-mask reuse."*
	- `amQi`:  *"Authors present a complete engineering solution, implementing a custom KV cache manager with... The system-level contributions, especially the fixed-size KV block design and swapping mechanism (Appendix G), address real implementation challenges that have prevented prior work from achieving practical speedups."*
- **Reviewers acknowledge the practical necessity and usefulness of our method/problem setting:**
    - `qBqd`:  *"Practical improvement to speculative decoding: single KV cache, no auxiliary draft model."*
    - `amQi`:  *"The problem and proposed approach are very timely and needed in the current scenario. The ideas and implementations allow us to realize the benefits of n-gram candidate-based decoding methods in a practical setting."*
- **Reviewers acknowledge the novelty and interestingness:**
	- `Cruo`:  *"Interesting idea: FAFO’s capability to perform n-gram drafting using a compressed KV cache and verify the n-gram drafts within the same forward pass is novel and technically elegant."*
	- `xKQY`:  *"The idea of combining KV-cache compression with lookahead decoding (draftless fumble decoding) looks novel and interesting."*

**We couldn't ask for more for this type of work.**

---

# **Reviewers' Concerns and Our Rebuttals**

A summary is only fair if it also faithfully highlights the concerns raised by our reviewers, so here we are. Beyond several factual clarifications, QAs, and simple additional ablation study / profiling requests, reviewers mainly expressed concerns in the following categories:

### **1. Batch inference support**


* `qBqd`:  *"The evaluation only reports results with a batch size of 1. How does FAFO perform under larger batch sizes or even with a multi-GPU system"*
 * `Cruo`:  *"Scalability under larger batches: Have you evaluated FAFO with larger batch sizes or multi-sequence decoding?"*

We have conducted additional profiling benchmarks according to the reviewers’ suggestions and find that **n-gram decoding is unlikely to be batch-friendly in its current form, and would require a work of its own to make it batch-capable.** We now explicitly recognize this shortcoming in the Limitation section.

That said, we note that efficient decoding work with a batched throughput focus usually requires drastically different designs (TriForce vs MagicDec, although both works are from the same lab). While FAFO is not batchable, its end-to-end performance still exceeds many batchable methods with batching enabled under many task scenarios.

---

> ### Author Response · Authors · 2025-12-04
> **Summary of reviewers' feedback and our rebuttals (2/2)**
>
> ### **2. Lack of algorithmic novelty**
>
>
> *  `qBqd` *"Limited novelty in algorithmic ideas: The core algorithmic concept is not particularly new — it essentially combines token sampling or KV-cache compression techniques with n-gram (Lookahead) decoding. The contribution lies more in integration than in conceptual innovation."*
> * `amQi` *"Limited Conceptual Novelty beyond existing Paradigms... While this is extremely non-trivial from an implementation standpoint, conceptually it does not stand out. The key idea that compressed caches could generate useful candidates has been explored in previous SSD contexts."*
>
>
> We don't disagree with the fact that our work does not provide much algorithmic novelty. **However, we argue algorithmic novelty is only one part of a contribution, where the context of a work's respective landscape matters.**
>
> Our work has been operating under many must-have constraints which, while favorable to end users, limit the freedom for algorithmic novelty (all prior art under such constraints lacks algorithmic innovation too). We also note that while the idea of lossy KV + n-gram is simple, nobody was able to get it to work for the past two years (since Lookahead Decoding's debut), suggesting our work is not a random repackaging but the identification of a good integration scheme that provides massive performance gains. Last, we want to quote our favorite reviewer guideline:
>
>
> > *“**The goal is to solve the problem, not to solve it in a complex way.** Simpler solutions are preferable — they are less brittle and easier to deploy.”*
>
> Many impactful efficiency works lack algorithmic novelty (e.g., QLoRA = standard quantization + LoRA; KIVI = per-channel quantization + sliding-window buffer), but they still end up achieving great community impact because they address pain points that no prior art did. Our work does the same for efficient decoding methods with a local deployment focus.
>
> ---
>
> ###  **3. Paper Writing**
>
> Our work received criticism on paper writing from `qBqd`, `amQi`, and `xKQY`. While at a glance, 3/4 reviewers agreement would suggest significant writing deficits, upon investigation, we find that these reviewers are actually requesting fixes that in conflict of each other.
>
> For instance, `qBqd` prefer us to include more system-level details, but this preference is in clear contrast to reviewer `amQi`'s feedback, who believes that if more system details are added, we should not submit to ICLR but to a more systems-focused venue. Similarly, **reviewer `Cruo` finds "clear presentation" to be a strength of our work** and appreciates how we thoroughly contrast FAFO to prior works. Yet, `qBqd` and  `amQi` generally want less background. Last, `amQi` might also have a significant misread of Section 1.2 of our work, where the reviewer-recommanded reformat is in fact exactly how we wrote in the first place.
>
> We list these conflicting examples not to discount the reviewer's feedback, but to  illustrate the point that **presentation is often a multifaceted subject with no universally good answer; *"one man's vulgarity is another's lyric"* is very much the name of the game given the growing size of ML community.** On this note, we ask for reviewer leniency and patience when encountering information ones are familiar with, as this is likely due to one's experience and expertise, while the general community (or another expert of different focuses) might find such basics useful.
>
> That said, we again agree that many improvements can be made in terms of organization. We have now:
>
> * Reduced Section 1.1 and moved failure mode details to the Appendix.
> * Added paragraph highlights in Section 1.2 to better distinguish background, criticism (of SSD), and motivation (for why n-gram).
> * Modified end-of-introduction contribution bullet points to better distinguish “contribution over SSD” and “contribution over Lookahead.”
> * Reduced repetition of the n-gram paradigm in Section 3.
> * Added a clearer roadmap in Section 3 leading up to the investigations in Sections 3.1 and 3.2.
> * Significantly expanded system-level details in Section 4.
> * Added a limitation section as Appendix B to highlight that FAFO currently lacks support for batched inference.
>
> We trust these updates would help.
>
> ---
>
> Although there is zero engagement, **we believe it is fair to argue that our work has addressed all major concerns in the best manner possible** — as most of them are rather factually oriented. We stand by the view that our proposed method can meaningfully contribute to the draftless + lossless efficient decoding field, while effectively reviving the stagnant n-gram paradigm by showcasing its promising potentials.
>
>
> We sincerely thank the ACs for the additional effort spent on our submission due to the leak, and we fully respect their discretionary judgment regardless of the outcome — we understand it is a tough job to navigate so many papers without active reviewer input, no matter how responsible an AC strives to be.

---

### Meta-Review · Area_Chair_ZqM3 · 2026-01-04

**Summary:**

Reviewer qBqd gave a score of 6 and viewed the paper as a solid technical contribution to speculative decoding with clear practical value. This reviewer appreciated the single KV cache, draftless design, and consistent latency improvements across models and tasks, as well as the engineering insights around attention sparsity and KV block management. The primary concern raised by this reviewer was the quality of presentation and organization. This reviewer found the early sections overly verbose and unfocused, with extensive repetition of background material on lossy KV cache compression that obscured the main ideas. In this reviewer’s view, the core innovations of FAFO were buried too deep in the paper, making it difficult for readers to quickly grasp the key contributions. This reviewer also noted that the algorithmic novelty was limited and that the contribution lay more in system integration than in new decoding principles. Additional concerns raised by this reviewer included insufficient system-level explanation of where the speedups originate and unclear scalability beyond batch size one, including CPU offloading overheads and behavior under larger batches or multi-GPU settings.

Reviewer Cruo gave a score of 6 and found the main idea technically elegant and clearly presented. This reviewer considered the integration of lossy KV cache compression with n-gram candidate generation and parallel verification within a single forward pass to be novel and well explained, and found the contrast with prior speculative decoding methods easy to follow. The main concerns raised by this reviewer focused on scalability rather than correctness or clarity. This reviewer noted that all experiments were limited to batch size one and questioned whether FAFO’s advantages would persist under larger batch sizes or multi-sequence decoding. This reviewer also pointed out that FAFO introduces additional computation per decoding step, which could shift the workload from memory-bound to compute-bound, potentially reducing speedups on larger batches or on GPUs with limited compute resources. This reviewer requested clearer quantification of the additional FLOPs introduced per step and a clearer characterization of when compute-bound behavior would dominate.

Reviewer amQi gave a score of 4 and acknowledged that the problem addressed by the paper is timely and important. This reviewer strongly valued the engineering contributions, particularly the custom KV cache manager, fixed-size KV block design, swapping mechanism, and integration with FlexAttention, which this reviewer saw as addressing real implementation barriers that have limited prior work. However, this reviewer expressed substantial concerns about the paper’s framing and positioning. This reviewer found the motivation and introduction confusing, especially the inconsistent use of the terms lossy and lossless, which conflated the use of lossy KV cache compression as a component with the claim of lossless end-to-end generation quality. This reviewer also felt that the transition from self-speculative decoding to n-gram decoding was abrupt and that the fundamental paradigm difference between sequential draft-then-verify and parallel draft-and-verify was introduced too late. In addition, this reviewer argued that many of the claimed benefits were inherent to the n-gram paradigm itself rather than unique contributions of FAFO, making some claims appear overstated. From a research perspective, this reviewer viewed the conceptual novelty as limited and felt that the work’s primary value lay in systems and engineering rather than in new algorithmic ideas.

Reviewer xKQY gave a score of 2 and recommended rejection. This reviewer’s strongest concern was the paper’s writing quality and structure, which this reviewer described as very verbose, poorly organized, and difficult to follow to the extent that this reviewer believed the technical contribution could not be fairly evaluated in its current form. This reviewer also questioned the choice of baseline for reported speedups, noting that while improvements over the original model were significant, the gains would likely be much smaller when compared against Lookahead, and that this distinction was not sufficiently emphasized. This reviewer further argued that it was unclear how much of the observed speedup should be attributed to FlexAttention rather than to FAFO itself. Additional concerns raised by this reviewer included the lack of ablation studies on lookback window size and missing related work such as LongSpec.

Overall, the reviews consistently highlighted weaknesses in presentation and clarity as a major issue, even among reviewers who were otherwise positive about the technical contribution. There was also broad agreement that the conceptual novelty is limited and that the primary contribution lies in system integration and engineering. Multiple reviewers additionally raised concerns about scalability beyond batch size one and about the potential transition from memory-bound to compute-bound behavior, which remains insufficiently explored.

**Reviewer Concerns:**

This meta-reviewer believes that several reviewer concerns were partially addressed in the rebuttal, but that a number of the most critical issues remain fundamentally unresolved.

On the positive side, this meta-reviewer acknowledges that the authors made a genuine effort to respond to presentation-related feedback. The rebuttal clarifies the intended distinction between lossy KV cache compression as an internal component and lossless end-to-end generation quality, and the authors report concrete structural changes to the paper, including reducing Section 1.1, improving signposting in the introduction, and more clearly separating contributions relative to self-speculative decoding and Lookahead. These responses partially address concerns raised by reviewers qBqd and amQi regarding terminology confusion and organization. The additional ablation studies requested by some reviewers, such as those on lookback window size and per-step FLOPs, also help clarify certain empirical behaviors of the method.

However, this meta-reviewer finds that the most substantive concerns raised by multiple reviewers were not convincingly resolved.

First, the issue of batch size and scalability remains largely unaddressed at a fundamental level. While the authors are transparent in acknowledging that FAFO is designed primarily for batch size one and latency-sensitive settings, this meta-reviewer views this limitation as structural rather than incidental. Multiple reviewers questioned whether the method’s advantages would persist under larger batch sizes, multi-sequence decoding, or more general serving scenarios, and the rebuttal effectively concedes that meaningful batched inference is not currently supported and may require a separate line of research. As a result, key questions about compute-bound behavior, throughput scalability, and applicability beyond narrow deployment regimes remain open. From this meta-reviewer’s perspective, these limitations significantly constrain the practical impact of the work and were not mitigated by the rebuttal.

Second, concerns regarding novelty were only partially addressed. The authors convincingly argue that FAFO operates under strict constraints and that integration-focused contributions can be valuable. This meta-reviewer agrees that the engineering effort is substantial and non-trivial. However, several reviewers, particularly amQi and qBqd, expressed concern that the core conceptual ideas largely reuse existing paradigms, namely lossy KV cache compression and n-gram or lookahead-style decoding, with the primary contribution lying in their combination and system-level realization. While the rebuttal defends this positioning philosophically, this meta-reviewer finds that it does not sufficiently articulate the inherent limitations of this integration-heavy approach, nor does it clearly delimit what new scientific insight is gained beyond improved execution. In particular, this meta-reviewer would have expected a more explicit discussion of the boundaries of the approach, including where and why the method may fail or be inherently constrained, rather than a predominantly defensive framing around novelty.

Third, some system-level attribution concerns remain unresolved. Although the authors argue that FlexAttention does not contribute to the reported speedups and is in fact slower than standard attention kernels, this meta-reviewer notes that questions about disentangling the contributions of architectural flexibility, kernel choice, and decoding logic remain difficult to evaluate conclusively. Similarly, while CPU offloading behavior and additional FLOPs are discussed in the rebuttal, these explanations reinforce the sense that the method trades one class of bottlenecks for another, without fully characterizing the resulting design space.

Overall, this meta-reviewer concludes that while the rebuttal addresses several surface-level issues and improves clarity, it does not sufficiently resolve the deeper concerns about scalability, general applicability, and the limited conceptual novelty of the approach. Given that the core strengths of the paper lie primarily in engineering integration rather than in new algorithmic insight, and that key limitations such as batch size remain intrinsic rather than incidental, this meta-reviewer believes that the outstanding concerns continue to outweigh the addressed ones, leading toward a negative recommendation.

**Reviewer Scores:**

This meta-reviewer believes that, while the rebuttal clarifies several points and demonstrates careful engagement with the feedback, it is unlikely to result in meaningful score increases. Some presentation-related concerns and terminology issues were partially addressed, and additional analyses helped clarify certain empirical behaviors. However, many of the remaining concerns are fundamental rather than superficial. In particular, the limitations around batch size and scalability were largely acknowledged rather than resolved, and the rebuttal confirms that the method is primarily designed for a narrow, latency-sensitive setting. In addition, concerns about limited conceptual novelty persist, as the core contribution remains an engineering integration of existing components rather than a clear algorithmic advance. As a result, although there may be some room for reconsideration at the margins, this meta-reviewer does not expect substantial upward score revisions following discussion.

---

### Decision · Program_Chairs · 2026-01-26

Reject